# MOTIONWEAVER: HOLISTIC 4D-ANCHORED FRAMEWORK FOR MULTI-HUMANOID IMAGE ANIMATION

**Xirui Hu**[1*], **Yanbo Ding**[2*], **Jiahao Wang**[1],
**Tingting Shi**[1], **Yali Wang**[2], **Guo Zhi Zhi**[3‡], **Weizhan Zhang**[1†],

[1]School of Computer Science and Technology, MOEKLINNS Lab,
 Xi'an Jiaotong University, Xi'an, 710049, China
[2]Shenzhen Institutes of Advanced Technology, Chinese Academy of Sciences, Shenzhen, China
[3]Institute of Artificial Intelligence (TeleAI), China Telecom, Beijing, China

{xiruihu,uguisu,shitingting}@stu.xjtu.edu.cn,
{yb.ding,yl.wang}@siat.ac.cn,
guozz2@chinatelecom.cn,
zhangwzh@xjtu.edu.cn

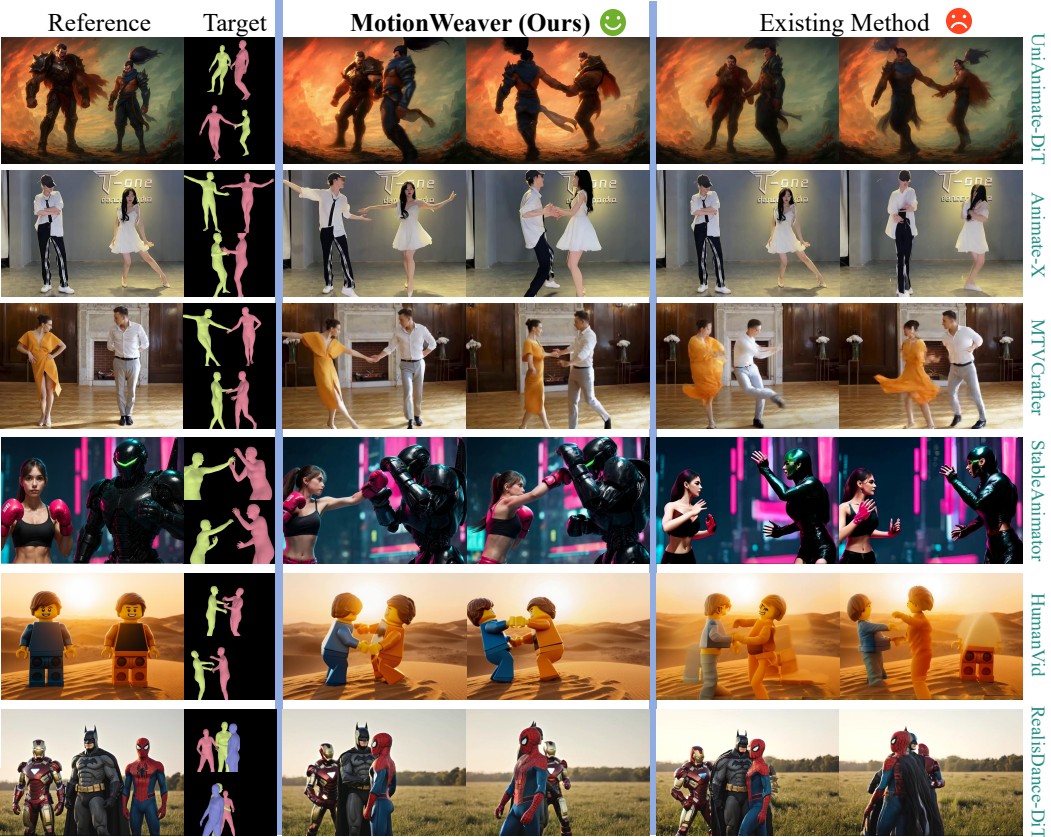

Figure 1: We propose MotionWeaver, a novel framework for **multi-humanoid image animation**, which effectively handles occlusions and complex interactions in multi-character scenarios, while showing strong **generalization** across diverse humanoid characters and artistic styles.

∗ These authors contributed equally; work done during internship at TeleAI.
† Corresponding author.
‡ Project leader.

## ABSTRACT

Character image animation, which synthesizes videos of reference characters driven by pose sequences, has advanced rapidly but remains largely limited to single-human settings. Existing methods struggle to generalize to multi-humanoid scenarios, which involve diverse humanoid forms, complex interactions, and frequent occlusions. We address this gap with two key innovations. First, we introduce unified motion representations that extract identity-agnostic motions and explicitly bind them to corresponding characters, enabling generalization across diverse humanoid forms and seamless extension to multi-humanoid scenarios. Second, we propose a holistic 4D-anchored paradigm that constructs a shared 4D space to fuse motion representations with video latents, and further reinforces this process with hierarchical 4D-level supervision to better handle interactions and occlusions. We instantiate these ideas in MotionWeaver, an end-to-end framework for multi-humanoid image animation. To support this setting, we curate a 46-hour dataset of multi-human videos with rich interactions, and construct a 300-video benchmark featuring paired humanoid characters. Quantitative and qualitative experiments demonstrate that MotionWeaver not only achieves state-of-the-art results on our benchmark but also generalizes effectively across diverse humanoid forms, complex interactions, and challenging multi-humanoid scenarios.

## 1 INTRODUCTION

Character image animation, which aims to synthesize videos of a reference character image driven by pose sequences estimated from an input video (Zhu et al., 2023; Peng et al., 2025; Chang et al., 2025), has gained significant attention due to its wide applications in film production, immersive content, and e-commerce (Lin et al., 2022; Lauer-Schmaltz et al., 2024; Islam et al., 2024; Song et al., 2025; Chamola et al., 2024; Qin & Hui, 2023).

While existing works have demonstrated promising results in motion control and character consistency (Gan et al., 2025; Xie et al., 2024; Tu et al., 2025b; Guo et al., 2024), they remain largely limited to single-human scenarios and fail to generalize to multi-humanoid settings characterized by diverse humanoid forms (e.g., robots, anthropomorphic animals, game avatars), rich interactions, and frequent occlusions. These limitations of current methods can be attributed to three primary factors: (1) **Deficient Motion Representations**: Existing methods inadequately disentangle motion from morphology, as control signals (e.g., skeleton maps or SMPL renderings) carry information about body proportions and shapes, which hinder generalization across diverse humanoid forms (Figure 1, fifth row). Moreover, they typically entangle control signals of multiple characters (e.g., multiple skeletons composited into one map), leading to identity confusion (Figure 1, top row), and limiting their applicability in multi-humanoid settings. (2) **Lack of Explicit 4D Modeling**: Prior approaches directly fuse video latents with control signals without explicitly modeling the spatiotemporal relationships among characters and with the scene (e.g., using naive cross-attention), thereby limiting their ability to capture complex multi-humanoid dynamics (Figure 1, third row). In particular, the absence of depth information is a critical drawback, making it difficult to resolve occlusions (Figure 3 and 1, bottom row) and to distinguish between a genuinely smaller body size and a greater camera distance. (3) **Suboptimal Training Strategy**: Current frameworks commonly rely on 2D pixel-wise MSE loss, which provides no explicit supervision for 4D motion and inadvertently couples motion with appearance, impairing motion understanding and leading to overfitting to human-specific visual features (Figure 1, fourth row). Collectively, these issues push conventional models to act as a mindless renderer of specified control signals rather than a real generator guided by motion information.

To address these limitations, we argue that models should be guided by more universal and scalable motion representations and trained within a thoroughly 4D-grounded perspective, thereby fostering a true grasp of motion dynamics to strengthen generalization. Guided by this principle, we introduce MotionWeaver, a framework built upon three innovations: (1) Unified-Choreography Core (UCC): extracts identity-agnostic motion signals and associates motion signals with corresponding driven characters to form unified motion representations, which robustly separate motion from morphology and enable applicability in multi-humanoid scenarios. (2) Hyper-Scene Integrator (HSI):

models a shared 4D space to fuse video latents and motion representations, enabling the resolution of frequent occlusions and dense interactions in multi-humanoid settings. (3) Hierarchical-4D Supervision (H4S): couples occlusion supervision at high-noise steps with motion-level supervision at low-noise steps, providing 4D motion supervision and mitigating overfitting to human appearance. Overall, the contribution of MotionWeaver lies in introducing novel **unified motion representations** and formulating a **holistic 4D-anchored paradigm** where motion extraction, motion–latent fusion, and training supervision are consistently grounded in 4D space. Building on these designs, MotionWeaver constitutes an end-to-end framework that supports multi-humanoid animation and robustly addresses interactions and occlusions.

To enhance the training of our model, we further curate a dataset containing 46 hours of multi-human videos, referred to as MultiHuman46, which features diverse interaction patterns and scenes. Additionally, we introduce DualDynamics, a benchmark of 300 videos, each showcasing two humanoid characters engaged in interaction-rich scenarios. These videos have undergone a rigorous filtering process to ensure quality. Quantitative and qualitative experiments demonstrate that MotionWeaver surpasses state-of-the-art methods, showcasing its generalization ability, identity preservation, and motion consistency in multi-humanoid scenarios. Our main contributions are summarized as follows:

- We propose MotionWeaver, a novel framework built upon the unified motion representations and the 4D-anchored paradigm, designed for multi-humanoid image animation involving diverse humanoid forms, rich interactions, and frequent occlusions.

- We introduce UCC to obtain unified motion representations, HSI and H4S to effectively construct a shared 4D space for fusing motion representations with video latents.

- We curate the MultiHuman46 dataset, which encompasses 46 hours of multi-human videos, and create DualDynamics, a benchmark comprising 300 videos of multiple humanoid characters in interaction-rich scenarios. Extensive experiments demonstrate that MotionWeaver surpasses state-of-the-art methods in multi-humanoid scenarios.

## 2 RELATED WORK

### 2.1 DIFFUSION TRANSFORMERS

Diffusion Transformers (DiTs) replace the traditional U-Net architecture with a Transformer model to denoise latent representations (Peebles & Xie, 2023; Xing et al., 2025). Recent advances combining DiTs with rectified flow techniques (Kong et al., 2024; Zheng et al., 2024; Yang et al., 2025; Blattmann et al., 2023) have achieved state-of-the-art generation quality. During inference, the model first establishes the overall structure, then progressively refines fine-grained details (Cao et al., 2024). Our training framework will be specifically designed to leverage this characteristic.

### 2.2 CHARACTER IMAGE ANIMATION

Recent research has shifted from GAN-based approaches (Siarohin et al., 2021a; 2018; Huang et al., 2021; Siarohin et al., 2021b) to diffusion-based methods (Chang et al., 2024; Wang et al., 2025a; Karras et al., 2023; Zhou et al., 2024; Ma et al., 2024a; Li et al., 2025; Xu et al., 2024a; Hong et al., 2025; Hu et al., 2025b) for image animation. Many methods employ skeleton maps to provide precise motion guidance (Zhang et al., 2025; Tu et al., 2024; Tan et al., 2025). Subsequent works further enrich the intermediate signals by incorporating mesh renderings (Zhu et al., 2024; Huang et al., 2024), dense pose maps (Wang et al., 2024a; Xu et al., 2024b; Yoon et al., 2024), facial expression controls (Luo et al., 2025; Ma et al., 2024b; Xu et al., 2025b), and hand controls (Pavlakos et al., 2024; Zhou et al., 2024). Further extensions include methods that integrate humans into given scenes to achieve scene affordance (Hu et al., 2025a; Kulal et al., 2023; Ostrek et al., 2024; Parihar et al., 2024), as well as techniques that model human–object interactions (Xu et al., 2025a; Men et al., 2025). Additional research explores camera embeddings for controllable viewpoints and camera motion (Shao et al., 2024; He et al., 2025; Hou & Chen, 2025; Yang et al., 2024b). Nevertheless, these approaches remain constrained to single-character scenarios and rely primarily on 2D control signals, which inevitably discard valuable cues from the real 4D world. Recent works have begun to incorporate 3D control signals. MTVCrafter (Ding et al., 2025) achieve lossless

information control by encoding 3D motion data. However, the inherent feature fusion mechanism and static RoPE encoding scheme are insufficient to address challenges specific to multi-character settings, such as position swapping and frequent occlusion. Efforts toward multi-character scenarios remain highly limited. While DanceTogether and Structural Video Diffusion (Chen et al., 2025; Wang et al., 2025c) tackle some of the challenges in multi-human scenes, they heavily rely on accurate human masks as control conditions. Although this approach effectively binds motion and corresponding identity, it requires strict alignment between reference characters and driving videos and cannot be readily extended to multi-humanoid settings. To address these limitations, we propose MotionWeaver, a framework that extracts more universal and scalable representations and is built entirely from a 4D perspective for animating multiple characters with diverse humanoid forms and complex interactions.

# 3  METHOD

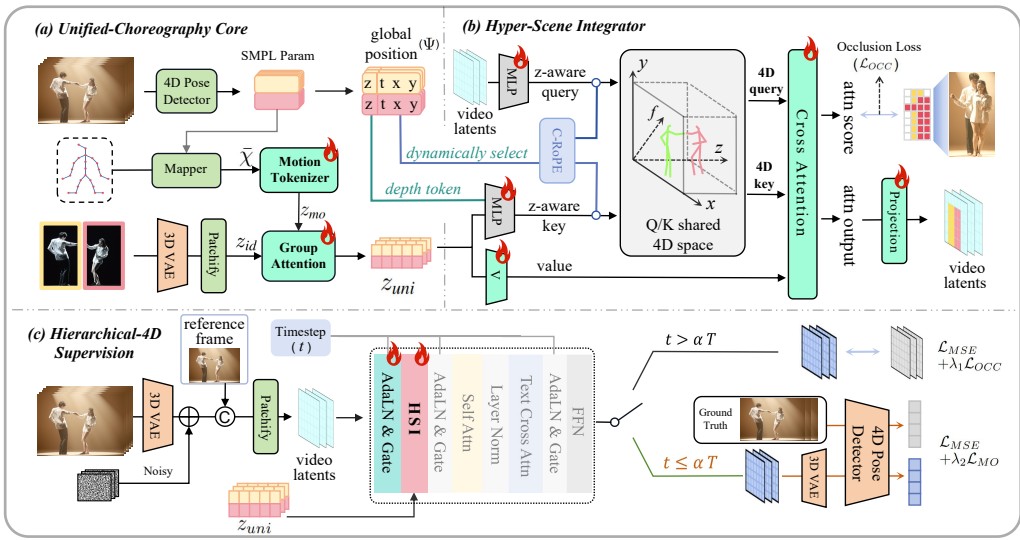

Figure 2: The overview of our MotionWeaver. (a) Unified-Choreography Core extracts unified motion representations ($z_{uni}$). (b) Hyper-Scene Integrator integrates the motion representations with video latents within a shared 4D space. (c) Hierarchical-4D Supervision utilizes timestep-specific 4D supervision to help the model effectively learn motion representations.

Our task is to animate **multiple humanoid** characters based on a reference image and a driving video containing desired poses while training exclusively on human datasets, which requires a more ingenious design. After presenting the preliminaries that underpin our approach (Section 3.1), our method revolves around two key contributions: (1) we introduce unified motion representations via the Unified-Choreography Core (Section 3.2), and (2) we establish a holistic 4D-anchored paradigm, which constructs a shared 4D space to integrate these 4D-informed motion representations into the base diffusion model through the Hyper-Scene Integrator (Section 3.3), while providing 4D-level guidance via our proposed Hierarchical-4D Supervision (Section 3.4). Finally, the training and evaluation are facilitated by our curated dataset, MultiHuman46, and the multi-humanoid benchmark, DualDynamics (Section 3.5).

## 3.1  PRELIMINARIES

**SMPL** (Skinned Multi-Person Linear) is a parametric 3D body model that represents the human surface as a mesh (Loper et al., 2015). It factorizes body variation into: (1) Shape parameters ($\beta$), which define individual body proportions and build; (2) Pose parameters ($\theta$), which represent relative joint rotations for the 24-joint kinematic skeleton; and (3) Translation parameters ($T$), which specify the global position in 3D space. By applying linear blend skinning and pose-dependent offsets

to a template mesh, SMPL provides a compact yet expressive representation for driving character motion.

**RoPE** (Rotary Positional Encoding) has been widely adopted in self-attention mechanisms (Su et al., 2024). Unlike traditional positional encoding, RoPE continuously rotates the position-dependent embeddings, allowing the model to preserve the relative distance between tokens more effectively. Mathematically, RoPE is represented as:

$$f_{\{q,k\}}(\boldsymbol{x}_m, m) = \boldsymbol{R}^d_{\Theta,m} \boldsymbol{W}_{\{q,k\}} \boldsymbol{x}_m, \tag{1}$$

where $\boldsymbol{x}_m$ represents the $m$-th token, $\boldsymbol{W}_{\{q,k\}}$ denotes the learnable projection matrices, and $\boldsymbol{R}^d_{\Theta,m} \in \mathbb{R}^{d\times d}$ is the rotary matrix corresponding to the $m$-th position. The rotary matrix is defined by pre-determined parameters $\Theta$.

## 3.2 UNIFIED-CHOREOGRAPHY CORE

To obtain the **unified motion representations** $z_{uni}$ that generalize across diverse humanoid forms and remain applicable to multi-humanoid scenarios, we introduce the Unified-Choreography Core (UCC), which extracts identity-agnostic motion tokens and associates them with the corresponding characters.

**Extract motion tokens for diverse humanoid forms generalization.** Unlike prior works that simply extract skeleton maps or SMPL renderings as motion representations, where appearance cues are inevitably entangled and thus impair generalization to diverse humanoid forms (Wang et al., 2025a; Zhu et al., 2024), we design a series of decoupling mechanisms to explicitly remove appearance information and distill identity-agnostic motion tokens. Specifically, we first employ a pose detector to extract the SMPL parameters of the characters in the driving video and convert them into joint coordinates, yielding $\chi \in \mathbb{R}^{P\times F\times J\times 3}$ as the raw representation. Here, $P$ and $F$ denote the number of persons and frames, respectively. To further eliminate appearance-related biases such as skeleton length and human-specific traits, we map these coordinates onto a standardized skeleton $\rho$, where fixed Euclidean distances between adjacent joints are enforced, resulting in standardized representations $\bar{\chi}$. In order to suppress frame-level noise and yield highly generalizable motion tokens $z_{mo}$, we design a task-specific tokenizer $\Phi_{tok}$, whose design is inspired by prior research (Ding et al., 2025). Specifically, $\Phi_{tok}$ downsamples $\bar{\chi}$ along the temporal axis to produce motion tokens. This overall procedure can be formally expressed as:

$$z_{mo} = \Phi_{tok}(\mathrm{Map}(\chi, \rho)) \in \mathbb{R}^{P\times(T\times J)\times D}, \tag{2}$$

where $T$ denotes the latent temporal length. Further information regarding the standardized skeleton $\rho$ can be found in Appendix G.

**Bind motion and character for multi-humanoid applicability.** Prior motion representations often lead to identity confusion in multi-humanoid settings, particularly when characters exchange positions. To address this issue, we explicitly bind motion to characters. Our approach builds upon a pre-trained I2V model, where reference frame features provide strong guidance throughout the generation process (Wan et al., 2025). Leveraging this property, we first segment the driven characters from the reference frame using masks, and then feed them into the base model's VAE and patchifying module to extract appearance features, yielding identity tokens $z_{id} \in \mathbb{R}^{P\times L\times D}$, where $L$ is the sequence length and $D$ is the feature dimensionality of each token. We then introduce a group attention module that associates the identity-agnostic motion tokens $z_{mo}$ with the corresponding identity tokens $z_{id}$, producing the unified motion representations $z_{uni} \in \mathbb{R}^{P\times(T\times J)\times D}$. For each character, the motion tokens serve as the query, while the identity tokens act as both the key and the value:

$$z_{uni[p]} = \mathrm{GroupAttn}(\mathrm{Q}(z_{mo[p]}), \mathrm{K}(z_{id[p]}), \mathrm{V}(z_{id[p]})) \in \mathbb{R}^{(T\times J)\times D}, \tag{3}$$

where $z_{uni[p]}$ denotes the unified motion representations for the $p$-th person. The group attention module is composed of a cross-attention layer followed by an MLP, both enclosed within residual connections.

## 3.3 HYPER-SCENE INTEGRATOR

While the unified motion representations provide reliable character-level signals, effectively fusing them with video latents in multi-humanoid settings is difficult due to dynamic spatiotemporal

positions, dense interactions, and occlusions. We argue that explicit 4D spatiotemporal modeling, particularly of depth cues, is essential to resolve these challenges. To this end, we propose the Hyper-Scene Integrator (HSI), which aligns and fuses motion representations and video latents within a shared 4D space to guide generation.

A critical aspect of this integration is specifying the 4D position of motion representations. Thus, we define a motion-unit $z_{uni[p,t]} \in \mathbb{R}^{J \times D}$, which represents the motion of character $p$ at latent timestep $t$. We then extract the SMPL translation parameters $(x, y, z)$ corresponding to this motion, compute their temporal average, and augment them with the latent timestep $t$. This yields $\Psi_{[p,t]} \in \mathbb{R}^4$, which specifies the global position of each motion-unit. To effectively construct a shared 4D space, we note that depth information, compared with the other three dimensions, is both crucial and more difficult for the model to capture, and that video latents inherently lack explicit depth cues. To this end, HSI introduces two complementary mechanisms: (1) a depth-aware attention that models the $z$-axis under occlusion-loss supervision, ensuring correct depth ordering and visibility; (2) a dynamic C-RoPE that models the axes $(t, x, y)$, providing direct positional cues that allow the model to rapidly capture spatiotemporal relationships.

**Depth-aware attention for axis** $z$**.** To encode depth information, we extract the corresponding $z$-coordinate from $\Psi_{[p,t]} \in \mathbb{R}^4$ for each motion-unit $z_{uni[p,t]} \in \mathbb{R}^{J \times D}$ and encode it with a sinusoidal embedding to obtain the depth token $z_{depth[p,t]} \in \mathbb{R}^D$. Each token of motion-unit is then concatenated with the depth token along the channel dimension, and processed by the $\mathrm{MLP_K}$ to generate a $z$-aware key. For each video latent $z_{v[t,x,y]}$, another $\mathrm{MLP_Q}$ is applied to obtain the $z$-aware query:

$$\bar{k}_{p,t,j} = \mathrm{MLP_K}([z_{uni[p,t,j]} \,\|\, z_{depth[p,t]}]) \in \mathbb{R}^D, \tag{4}$$

$$\bar{q}_{t,x,y} = \mathrm{MLP_Q}(z_{v[t,x,y]}) \in \mathbb{R}^D. \tag{5}$$

To supervise the learning of the depth-aware MLPs, we first obtain the final attention score matrix $\mathbf{H} \in [0,1]^{(THW) \times (PTJ)}$ from the cross-attention module. For the $p$-th character, we take the corresponding slice along the $P$ dimension and sum over the last two dimensions ($T$ and $J$), producing a character-specific map $\mathbf{h}_p \in [0,1]^{(THW)}$. Collecting the maps of all characters yields $\{\mathbf{h}_i\}_{i=0}^{P-1} \in [0,1]^{P \times (THW)}$. Then, we construct ground-truth masks $\{\mathbf{m}_i\}_{i=0}^{P-1} \in \{0,1\}^{P \times (THW)}$. In regions where characters overlap, the mask value is enforced to be zero for the character that lies farther along the depth axis, thereby explicitly encoding the occlusion relationship. To ensure accurate depth learning, we minimize the following occlusion loss:

$$\mathcal{L}_{\mathrm{OCC}} = \frac{1}{TP} \sum_{i=1}^{P} \mathrm{MSE}(\mathbf{h}_i, \mathbf{m}_i), \tag{6}$$

which enforces the model to learn the depth-aware occlusion relationships correctly.

**Dynamic C-RoPE for axes** $(t, x, y)$**.** To effectively construct a shared space for modeling positional relationships among characters and with the scene, we first build the corresponding rotary matrix set for dynamic C-RoPE (Cross-Attention Shared RoPE), denoted as $\phi \in \mathbb{R}^{(T \times H \times W) \times (D \times D)}$. The rotary matrix at any point $(t, x, y)$ is given by:

$$\phi_{[t,x,y]} = \tilde{\boldsymbol{R}}_{\Theta,t,x,y}^d = \begin{pmatrix} \boldsymbol{R}_{\Theta,t}^{d/3} & \mathbf{0} & \mathbf{0} \\ \mathbf{0} & \boldsymbol{R}_{\Theta,x}^{d/3} & \mathbf{0} \\ \mathbf{0} & \mathbf{0} & \boldsymbol{R}_{\Theta,y}^{d/3} \end{pmatrix}, \tag{7}$$

where $\boldsymbol{R}_{\Theta,i}^{d/3} \in \mathbb{R}^{\frac{d}{3} \times \frac{d}{3}}$ represents the rotary matrix associated with the $i$-th position in the respective dimension. Since the $(x, y)$ indices of frame latents reside in the pixel space, whereas $(x, y) \in \Psi$ are defined in the camera space, we further project $(x, y) \in \Psi$ onto the same perspective plane for alignment. Subsequently, the rotary matrix is dynamically selected from $\phi$ based on the global position of each motion-unit to rotate the key, while the query is rotated using its associated rotary matrix:

$$\tilde{k}_{p,t,j} = \tilde{\boldsymbol{R}}_{\Theta,\Psi[p,t]}^d \bar{k}_{p,t,j}, \tag{8}$$

$$\tilde{q}_{t,x,y} = \tilde{\boldsymbol{R}}_{\Theta,t,x,y}^d \bar{q}_{t,x,y}. \tag{9}$$

To further clarify, we provide a code snippet in Appendix N. The final step involves projecting the unified motion representation $z_{uni}$ to obtain the corresponding value, after which cross-attention is applied to fuse the video latents with the unified motion representations within a shared 4D space.

## 3.4 HIERARCHICAL-4D SUPERVISION

Since HSI has already established a 4D space for integrating 4D-informed motion representations, we further introduce Hierarchical-4D Supervision (H4S) to regulate the model's learning of occlusions and interactions, thereby establishing a holistic 4D-anchored paradigm. H4S provides 4D motion supervision decoupled from appearance to enhance the efficiency of motion learning and reduce overfitting to human appearance. This design is also aligned with the inherent characteristics of diffusion models, which initially learn global layout and structure during the early, high-noise denoising steps before progressively focusing on finer details in the later, low-noise steps (Yang et al., 2024a; Chen et al., 2024). During training, H4S utilizes a dynamic loss function that adapts to the current noise level, which is defined as follows:

$$\mathcal{L}_{H4S} = \left\{ \begin{array}{ll} \mathcal{L}_{MSE} + \lambda_1 \mathcal{L}_{OCC}, & t \geq \alpha T \\ \mathcal{L}_{MSE} + \lambda_2 \mathcal{L}_{MO}, & t < \alpha T \end{array} \right. \tag{10}$$

where $t$ represents the current timestep sampled during training, with $T$ being the total number of timesteps. The loss function switches between auxiliary components based on a predefined threshold $\alpha$. For high-noise timesteps, we apply a standard Mean Squared Error (MSE) loss $\mathcal{L}_{MSE}$, in conjunction with the **occlusion loss** $\mathcal{L}_{OCC}$ in Eq. (6), which ensures that the model establishes accurate occlusion relationships between characters from the very beginning of the denoising process. For low-noise timesteps, the occlusion loss is replaced with a **motion-level loss** $\mathcal{L}_{MO}$. At this stage, the model can already predict the denoised latents with near-accurate precision. These denoised latents are subsequently decoded by the VAE and passed through the stem of a pre-trained 4D pose detector. The extracted motion features are then aligned with their ground-truth using an MSE loss, forming $\mathcal{L}_{MO}$, which transfers strong 4D motion priors into our HSI. Further details on the extraction of motion features are provided in Appendix E. In addition, we introduce trainable timestep-conditioned AdaLN and gating mechanisms for modulating HSI, which endow the model with stronger timestep sensitivity and finer control over the 4D feature fusion process.

## 3.5 MULTIHUMAN46 DATASET AND DUALDYNAMICS BENCHMARK

To facilitate the training, we curate MultiHuman46, a 46-hour multi-human video dataset that encompasses a wide range of interaction patterns and scenes (e.g., boxing, fencing, and dancing). The clips are collected from public datasets, web-crawled materials, and carefully screened AI-generated videos. We estimate SMPL parameters using a state-of-the-art multi-person pose detector (Newell et al., 2025) and apply a multi-stage filtering pipeline to retain only video clips that exhibit strong temporal consistency, high motion fidelity, and reliable SMPL parameter accuracy.

Given that publicly available character animation benchmarks primarily focus on single-human settings, such as Fashion (Zablotskaia et al., 2019) and TikTok (Jafarian & Park, 2021), we aim to fill the gap in benchmarks for multi-humanoid image animation. To this end, we construct DualDynamics, a novel benchmark comprising 300 videos, each depicting interactions between two humanoid characters. DualDynamics is purposefully constructed to rigorously assess models' understanding of motion under diverse humanoid forms and complex interactions. All benchmark videos are produced by a professional animation team and undergo filtering to ensure rich character interactions. Further details on the dataset and the benchmark are provided in Appendix I.

## 4 EXPERIMENTAL RESULTS

### 4.1 EXPERIMENTAL SETUP

We employ CoMotion (Newell et al., 2025) to estimate the SMPL parameters of each individual, and further use these parameters to construct the corresponding masks. As a state-of-the-art method for multi-person pose detection under frequent occlusions and interactions, CoMotion provides robust depth and pose predictions that yield reliable motion signals for our pipeline. We first train a robust 4D tokenizer. We then adopt Wan2.1-I2V-14B-480P (Wan et al., 2025) as the base model and insert our HSI into it at regular intervals of every four blocks. At this stage, all parameters are kept frozen, while only our group attention module and HSI are trainable. We set $\lambda_1 = 1$ and $\lambda_2 = 1$, and choose $\alpha = 0.6$, with the rationale provided in Appendix H. The input video clips consist of 49 consecutive frames. Optimization is performed using the AdamW optimizer with $\beta_1 = 0.9$, $\beta_2 = 0.99$, a weight

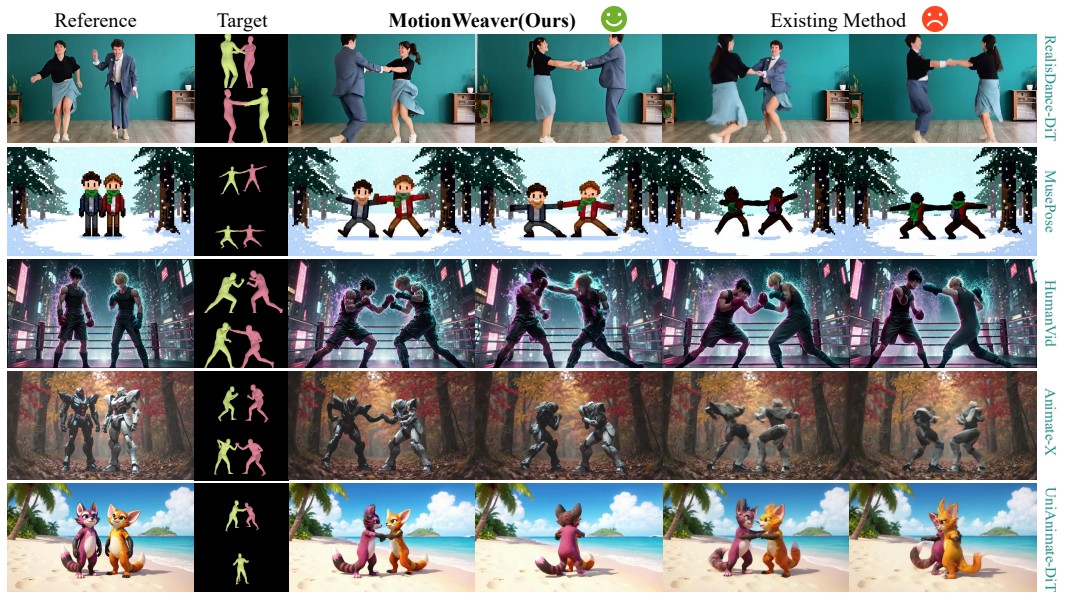

Figure 3: Qualitative Comparison with Existing Methods. The *yellow* and **red** meshes indicate the target motions of the *left* and **right** characters from the reference image, respectively. Our MotionWeaver method achieves superior identity preservation and motion accuracy for multiple humanoid characters. Notably, it is the only approach that correctly handles dense inter-character interactions and occlusions.

decay of $1 \times 10^{-2}$, and a learning rate of $2 \times 10^{-5}$. Training is conducted on 8 NVIDIA H100 80G GPUs with a batch size of 1 per GPU. To reduce memory consumption and enable more efficient training, we present a set of tailored training strategies, detailed in Appendix J.

## 4.2 SOTA COMPARISON

**Baselines.** We evaluate our method with leading and open-source baselines, including RealisDance-DiT (Zhou et al., 2025), UniAnimate-DiT (Wang et al., 2025b), Animate-X (Tan et al., 2025), HumanVid (Wang et al., 2024b), StableAnimator (Tu et al., 2025a), MusePose(Tong et al., 2024) and MimicMotion (Zhang et al., 2025).

**Quantitative Evaluation**. The results are reported in Table 1, covering nine widely used metrics: L1, PSNR (Horé & Ziou, 2010), SSIM (Wang et al., 2004), LPIPS (Zhang et al., 2018), DISTS (Ding et al., 2022), CLIP (Hessel et al., 2021), FID (Heusel et al., 2017), FID-VID (Balaji et al., 2019), and FVD (Unterthiner et al., 2019). On the DualDynamics benchmark, where each video is composed of two humanoid characters, MotionWeaver consistently outperforms all the baselines across all the metrics. This clearly demonstrates the superiority of our approach in multi-humanoid scenarios characterized by rich interactions, frequent occlusions, and diverse humanoid forms.

**Qualitative Evaluation**. The comparisons are provided in Figures 1 and 3, with additional qualitative results available in Appendix P and the supplementary materials. The design of Animate-X is focused on single-character scenarios, significantly limiting its applicability to multi-character contexts. Though methods such as MimicMotion, MusePose, StableAnimator, and HumanVid can be extended to multi-human settings, they fail to generalize well across diverse humanoid forms. UniAnimate-DiT and RealisDance-DiT exhibit only limited generalization and struggle to handle occlusions (Figure 3, top and bottom rows) and identity confusion (Figure 1, top row) in multi-humanoid scenarios. In contrast, our approach demonstrates clear superiority in multi-humanoid settings, effectively addressing dense interactions, frequent occlusions, and diverse humanoid forms. Further results also highlight (1) its competitive performance in **real-world scenarios** (Figure 1 second row, Figure 3 top row, and Appendix), and (2) its robust generalization to **settings involving more than two characters** (Figure 1 bottom row and Appendix). Both qualitative and quantitative evaluations consistently demonstrate the effectiveness of our unified motion representations and the

Table 1: Quantitative results on the DualDynamics benchmark. Our method consistently outperforms all baselines across all metrics, demonstrating significant advantages in multi-humanoid scenarios involving diverse humanoid forms and rich interactions.

| Method | Video Metrics | | Image Metrics | | | | | |
|---|---|---|---|---|---|---|---|---|
| | FVD ↓ | FID-VID ↓ | FID ↓ | L1 ↓ | PSNR ↑ | SSIM ↑ | LPIPS ↓ | CLIP ↑ |
| MimicMotion | 312.4 | 74.23 | 71.72 | 0.6098 | 26.04 | 0.5165 | 0.4319 | 0.7842 |
| MusePose | 298.3 | 59.14 | 60.46 | 0.5741 | 28.12 | 0.5301 | 0.3416 | 0.8109 |
| StableAnimator | 262.5 | 35.19 | 34.97 | 0.5261 | 27.11 | 0.5341 | 0.3721 | 0.7741 |
| Animate-X | 230.1 | 32.47 | 30.22 | 0.5361 | 28.15 | 0.5276 | 0.3780 | 0.8539 |
| HumanVid | 174.6 | 29.12 | 31.58 | 0.5122 | 27.03 | 0.4576 | 0.4197 | 0.8214 |
| UniAnimate-DiT | 172.3 | 24.98 | 22.87 | 0.5743 | 29.11 | 0.5399 | 0.3482 | 0.8601 |
| RealisDance-DiT | 164.6 | 22.12 | 23.26 | 0.5341 | 29.06 | 0.5216 | 0.3271 | 0.8813 |
| **Ours** | **145.7** | **20.34** | **19.41** | **0.4836** | **29.19** | **0.5428** | **0.3213** | **0.9041** |

Table 2: Ablation study of the core components on the DualDynamics benchmark. The complete design achieves the best overall performance.

| Method | Video Metrics | | Image Metrics | | | | | |
|---|---|---|---|---|---|---|---|---|
| | FVD ↓ | FID-VID ↓ | FID ↓ | L1 ↓ | PSNR ↑ | SSIM ↑ | LPIPS ↓ | CLIP ↑ |
| Ours w/o MNP | 198.5 | 27.28 | 25.29 | 0.5524 | 26.34 | 0.519 | 0.3751 | 0.7801 |
| Ours w/o GAM | 183.2 | 25.76 | 24.45 | 0.5343 | 27.98 | 0.537 | 0.3396 | 0.8941 |
| Ours w/o DAA | 167.1 | 21.80 | 22.03 | 0.5252 | 28.64 | 0.522 | 0.3452 | 0.8921 |
| Ours w/o DCR | 225.6 | 27.31 | 25.29 | 0.5541 | 26.28 | 0.511 | 0.3657 | 0.8270 |
| Ours w/o H4S | 174.3 | 24.22 | 21.46 | 0.5011 | 29.04 | 0.524 | 0.3429 | 0.8714 |
| **Ours** | **145.7** | **20.34** | **19.41** | **0.4836** | **29.19** | **0.5428** | **0.3213** | **0.9041** |

holistic 4D-anchored paradigm in integrating these representations into the base diffusion model to drive character animation.

### 4.3 ABLATION STUDY

We conducted an ablation study to verify the necessity of our core components: Unified-Choreography Core, Hyper-Scene Integrator, and Hierarchical-4D Supervision. As shown in Table 2, the results on the DualDynamics benchmark demonstrate the performance impact when these elements are individually altered or removed. Additionally, we provide qualitative visual experiments in Figure 4 to visually illustrate the contribution of each module.

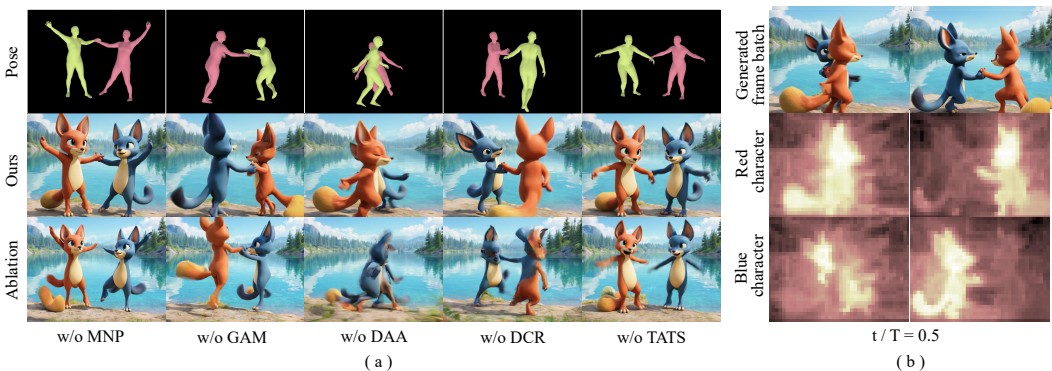

Figure 4: Visual Comparison of Ablation Results. (a) The *yellow* and **red** meshes represent the target motions of the *red* and **blue** characters, respectively. The original design achieves the best visual performance among all variants. (b) Visualization of attention maps between frame latents and per-character unified motion representations.

**Unified-Choreography Core.** (1) *Motion Normalization Pipeline (MNP)*. Directly converting raw SMPL parameters into joint coordinates and encoding them through a linear layer degrades the animation quality, especially when the reference character's physical attributes deviate from the motion source, leading to unrealistic limb configurations. (2) *Group Attention Module (GAM)*. Removing the group attention module and directly injecting motion tokens into HSI leads to incorrect motion-character assignments, particularly when characters swap positions.

**Hyper-Scene Integrator.** (1) *Depth-Aware Attention (DAA)*. Removing the use of depth tokens and occlusion-loss supervision, the model could not correctly handle occlusions between characters. (2) *Dynamic C-RoPE (DCR)*. In the absence of dynamic C-RoPE, we observe unstable training behavior and slower convergence, which ultimately causes a marked deterioration in animation quality. (3) Furthermore, we visualize the *attention maps* between frame latents and per-character unified motion representations at $t/T = 0.5$. These visualizations clearly demonstrate that Hyper-Scene Integrator successfully captures the underlying 4D spatiotemporal structure, thereby enhancing the interactive fusion of video latents with the motion representations.

**Hierarchical-4D Supervision.** When the training objective is restricted to a fixed combination of MSE and occlusion loss, the model exhibits degraded control over humanoid poses, often resulting in blurry and temporally inconsistent motion renderings.

**Scalability.** Qualitative experiments in 5 demonstrate that our method can be generalized to scenarios with more than two characters during the inference stage, which validates the superiority of our holistic 4D-anchored paradigm. While the training phase incorporates sophisticated hierarchical supervision, the inference pipeline remains elegant and efficient: the UCC extracts unified motion representations, which are then integrated by the HSI to provide precise, depth-aware guidance for video synthesis.

Target          Generated          Reference          Target          Generated

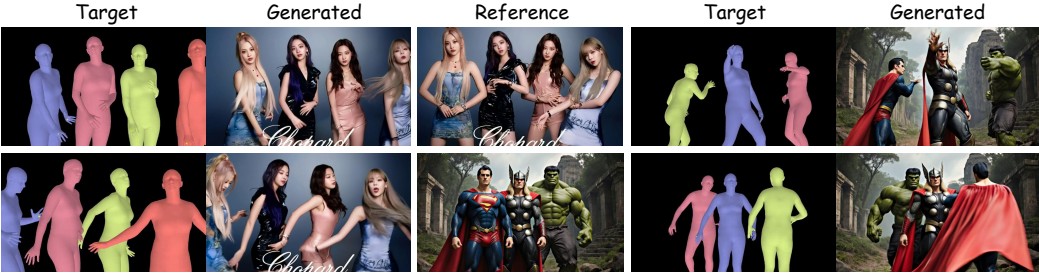

Figure 5: MotionWeaver can indeed process scenes with more than two humanoids.

## 5  CONCLUSION

We have presented MotionWeaver, a holistic 4D-anchored framework designed for multi-humanoid animation. By introducing the Unified-Choreography Core (UCC), we explicitly bind identity-agnostic motion to character-specific appearance features, effectively eliminating identity confusion during complex interactions. The Hyper-Scene Integrator (HSI) and Hierarchical-4D Supervision (H4S) further unify these representations within a shared 4D space, providing the depth-aware guidance necessary to handle frequent occlusions. Supported by our MultiHuman46 dataset and DualDynamics benchmark, MotionWeaver achieves state-of-the-art performance and demonstrates strong generalization across diverse interaction patterns. Our work establishes a robust foundation for the holistic modeling of multi-character dynamics in video synthesis.

### ACKNOWLEDGMENTS

This work was supported in part by NSFC under Grant 62192781, 62137002, Research Project Funded by the Key Research and Development Project in Shaanxi Province 2022GXLH-01-03, and the Project of China Knowledge Centre for Engineering Science and Technology.

## 6 ETHICS STATEMENT

While our framework is intended purely for academic research, we recognize that its ability to synthesize controllable videos of real individuals could potentially be misused for generating deceptive or harmful content. We explicitly discourage such misuse and stress that applications must adhere to ethical standards and legal regulations. Furthermore, our benchmark and methodology emphasize diversity in scenarios and humanoid forms to avoid bias toward specific identities. This research fully complies with the ICLR Code of Ethics.

## 7 REPRODUCIBILITY STATEMENT

We have made substantial efforts to ensure the reproducibility of our work. The overall model design is thoroughly documented: Section 3.2 and Section 3.3 provide detailed descriptions of the model architecture, while Section 3.4 elaborates on the training strategies employed. Additional settings, including hyperparameters, optimization choices, and detailed implementation of the methods, are presented in Section 4.1 and further expanded in the Appendix. To support reproducibility of the experimental pipeline, we provide a comprehensive description of the data collection process, preprocessing steps, and benchmark construction in Appendix I. In addition, we release part of the constructed benchmark as well as the key code components necessary to reproduce the main experiments in the supplementary materials. Together, these resources ensure that readers can both understand the methodological details and reliably reproduce our reported results.

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

# Appendix

Table of Contents

## A    ABLATION ON 4D-ANCHORED FRAMEWORK

We selected the two strongest state-of-the-art methods, UniAnimate-DiT and RealisDance-DiT, for this comparative analysis. These models were subsequently fine-tuned on our specialized multi-human dataset, MultiHuman46. As demonstrated in 6, these baselines remain unable to robustly handle the core challenges inherent in multi-humanoid animation, including diverse humanoid forms, complex interactions, and frequent occlusions, even after being trained on task-specific data.

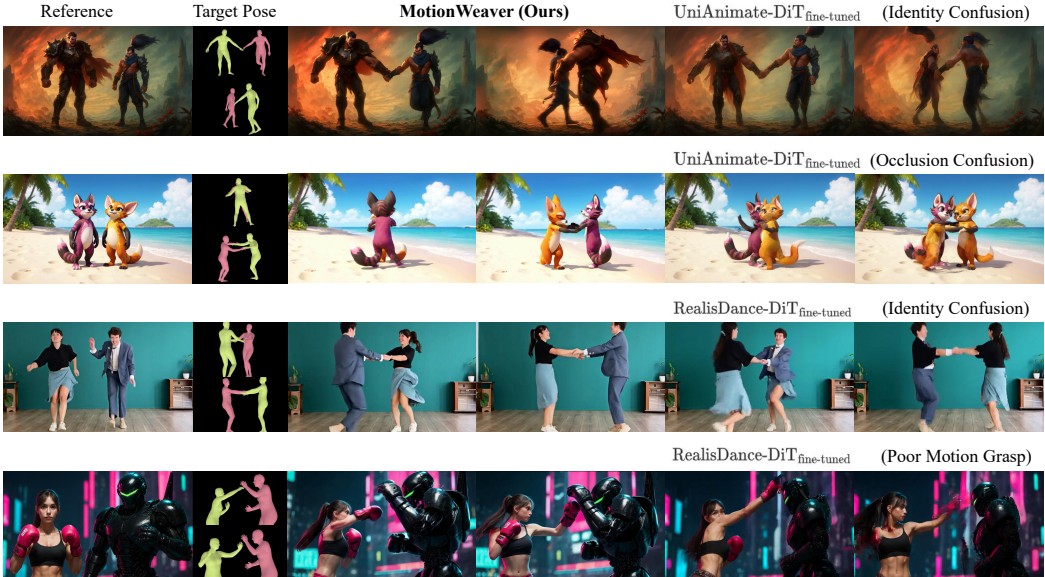

Figure 6: After fine-tuning baselines on our specialized multi-human dataset, they still fail to robustly address the core challenges inherent to multi-humanoid animation.

## B    STANDARD SINGLE-PERSON ANIMATION TASK

We conducted qualitative experiments on the standard single-person TikTok benchmarks in 7. Our approach achieves competitive performance in standard single-person animation tasks, unequivocally demonstrating no performance degradation compared to state-of-the-art specialized methods.

## C    EVALUATION ON REAL-CAPTURE SEQUENCES

We present qualitative results in 8. We observe that our method still effectively suppresses identity drift even in this challenging scenarios, which further demonstrates the robustness of our approach. The experiments utilized 81-frame video sequences, corresponding to the maximum sequence length employed during the training of the base model.

## D    THE USE OF LARGE LANGUAGE MODELS

In preparing this paper, large language models (LLMs) were used only as general-purpose tools for improving the clarity and fluency of the writing. They did not contribute to research ideation, methodological design, data collection, or experimental analysis. All content was reviewed and revised by the authors, who take full responsibility for the final manuscript.

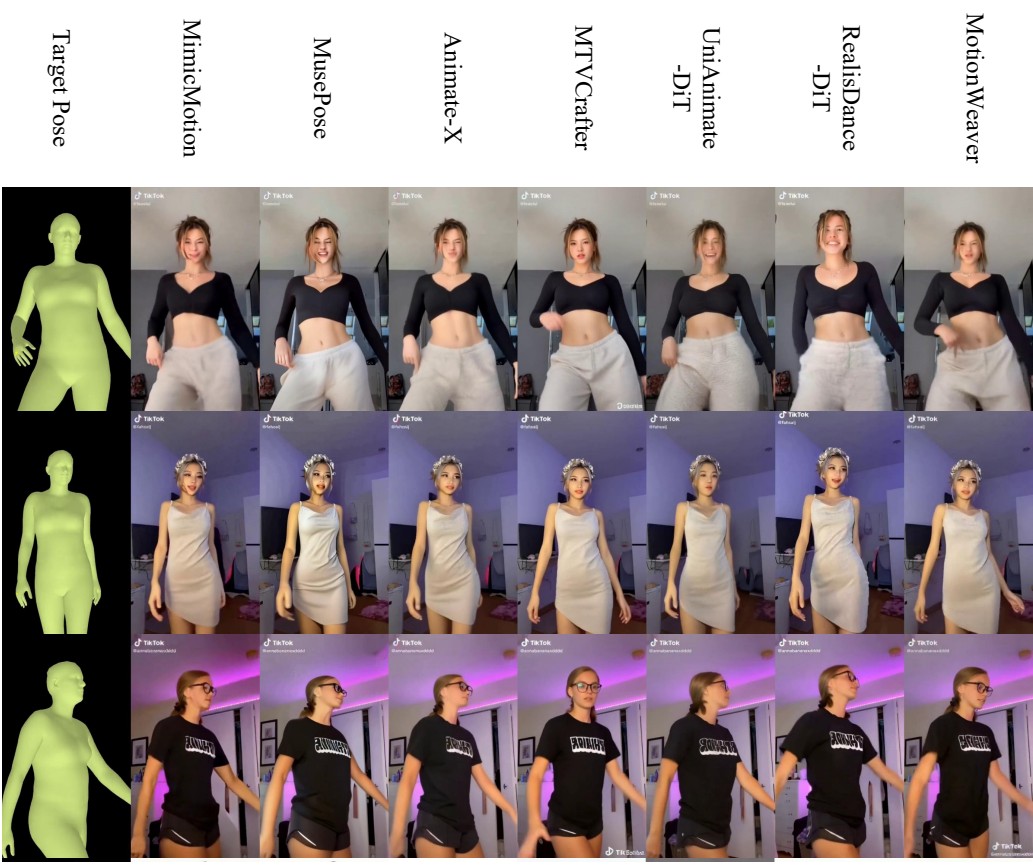

Figure 7: Qualitative experiments on TikTok datasets. Our approach is also capable of achieving competitive results in the single-human setting, **which has been the focus of prior work**.

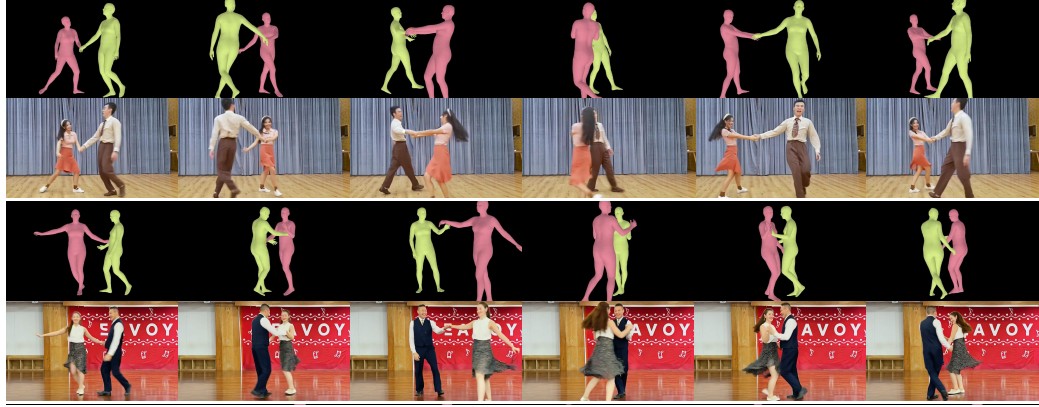

Figure 8: Long-horizon real-capture sequence tests with frequent position exchanges.

# E    EXTRACTION OF MOTION FEATURES

We leverage the intermediate representations of the state-of-the-art multi-person pose estimation model CoMotion as our motion features. Specifically, CoMotion first employs an image backbone to encode the input image, followed by a detection head that produces three groups of multi-scale feature maps with dimensions (1,1024,16,16), (1,1024,8,8), and (1,1024,4,4). These feature maps are then processed by three decoders, yielding a total of 1344 candidate detections. A subsequent

non-maximum suppression step is applied to filter redundant predictions and retain the final pose estimates. Owing to CoMotion's strong performance in robust pose estimation, we argue that its intermediate representations effectively capture motion-relevant information. We therefore adopt these intermediate features as motion signals to supervise the training of our model. However, since extracting motion features requires passing them through a VAE decoder, which introduces additional GPU memory overhead, we restrict the extraction to the first two latents of each input sequence.

## F    EXTRACTION OF APPEARANCE FEATURES

Our motivation originates from the observation in Wan2.1-I2V-14B-480P that the features extracted from reference frames through the VAE encoder followed by the Patchifier play a decisive role in guiding video generation. These features effectively encode the visual appearance of the reference characters. By isolating and reusing such features, we can explicitly associate each character with its corresponding motion signal. This binding is particularly crucial in multi-character scenarios, as it enables the model to handle occlusions and inter-character interactions while preserving identity consistency.

In the original Wan2.1-I2V-14B-480P workflow, the processing pipeline operates as follows. A reference frame is first padded with blank frames to match the length of the target video sequence. The padded sequence is then passed through the VAE encoder to obtain latent features. These features are concatenated along the channel dimension with two additional components: (1) a latent mask and (2) noise latents. The combined representation is processed by the Patchifier to produce tokens that serve as inputs for the diffusion model.

To remain consistent with the base model while ensuring that our extracted features align with its representational design, we propose the following procedure for extracting appearance features, as shown in Figure 9. We begin by masking out the character from the reference frame to construct a new image. This image is then treated as a single frame and passed through the VAE encoder to obtain the latent that captures fine-grained character appearance. We then duplicate the latent once to approximate noise latents and construct an all-ones matrix as a surrogate for the latent mask. The three components, namely the latent, the duplicated "pseudo-noise" latent, and the all-ones mask, are concatenated along the channel dimension and passed through the Patchifier. The resulting output constitutes the appearance features, which explicitly encode character-specific visual cues while remaining aligned with the feature representation of the base model. This design creates the possibility of reliably binding motion signals to the correct identities, thereby enabling robust performance in complex multi-character video generation scenarios.

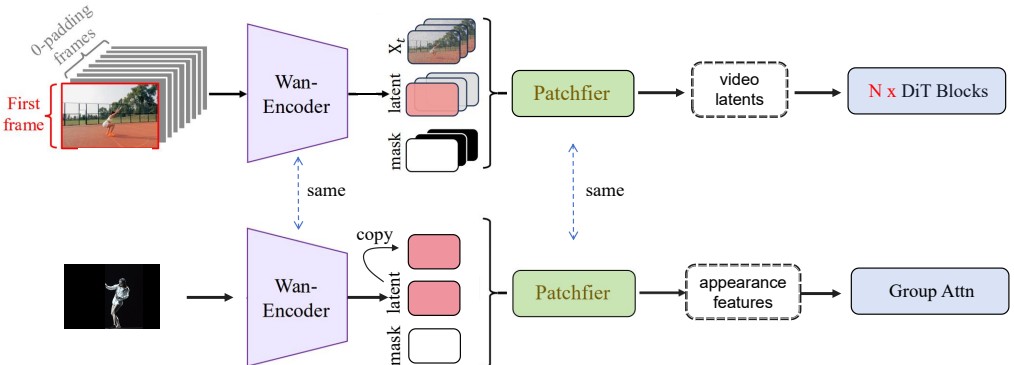

Figure 9: The upper figure illustrates the feature extraction pipeline of the Wan base model, while the lower figure presents our appearance feature extraction process, which is deliberately designed to align as closely as possible with the base model.

## G    RECONSTRUCTION OPERATOR

Considering that SMPL is inherently a human-centered model, the joint coordinates derived from SMPL parameters inevitably carry biases that are tightly coupled with human body proportions. Such biases may hinder the generalization of downstream motion representation, especially when extending animation to humanoid characters whose shapes, limb ratios, or stylistic designs deviate significantly from the human norm. To mitigate this limitation, we reconstruct a standardized skeleton by enforcing a fixed Euclidean distance between each pair of adjacent joints, thereby decoupling the structural representation from specific body proportions. This process results in a uniform skeletal topology that emphasizes the relational geometry of joints rather than their absolute scales. By doing so, the resulting skeleton provides a more robust and identity-agnostic foundation for motion encoding across diverse humanoid characters. A visualization of our standardized skeleton is presented in Figure 10, where the normalized structure highlights consistent spatial relationships while eliminating distortions introduced by body-shape variability.

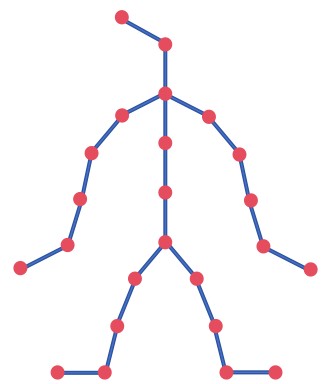

Figure 10: Visualization of our standardized skeleton.

## H    RATIONALE FOR HYPERPARAMETER SELECTION IN H4S

We conducted extensive qualitative experiments to investigate the behavior of Wan2.1-I2V-14B-480P during inference across different timesteps. The experiment was designed as follows: given a specific timestep and its corresponding noisy latent, the model is tasked with estimating the velocity vector field. This estimated field, together with the noisy latent, is then used to directly compute the denoised latent, which is subsequently decoded back into the pixel space. A representative set of results is presented in Figure 11.

Our observations confirm that the model exhibits the canonical characteristics of diffusion models: in the early denoising stages, the model focuses primarily on capturing the global structure and layout of the scene, while in the later stages, it progressively refines fine-grained details. Notably, we find that around $t/T = 40\%$, Wan2.1-I2V-14B-480P begins to predict an almost accurate velocity field, producing relatively clear video frames and showing increasing attention to motion details. Based on this behavior, we empirically select $\alpha = 0.6$ as the optimal setting.

## I    MULTIHUMAN46 DATASET AND DUALDYNAMICS BENCHMARK

### I.1    MULTIHUMAN46 DATASET

We construct our MultiHuman46 dataset through a carefully designed multi-stage curation pipeline, which integrates heterogeneous data sources and applies rigorous filtering to guarantee both diversity and quality. The overall process comprises four stages: video collection, clip segmentation, automatic quality filtering, and manual validation.

**Video collection**. To maximize diversity in both appearance and motion, we draw upon three complementary sources of videos. First, a portion of the data comes from publicly available datasets, including TikTokDataset (Jafarian & Park, 2021), HumanVidDataset (Wang et al., 2024b), Swing Dance (Maluleke et al., 2024), and CHI3D (Fieraru et al., 2023), which are widely adopted for human motion research. Second, we collect web-crawled videos exclusively from publicly accessible materials, ensuring that all samples originate from open platforms. Third, to further broaden the diversity of scenarios, we incorporate AI-generated videos. Specifically, we first employ the FLUX.1 Krea model (Greenberg, 2025) to generate still images depicting 2–3 humans standing in diverse scenes, with scene descriptions specified in prompts (see Section Q). To ensure reliability, we use the CoMotion model to automatically verify that the number of generated humans matches the

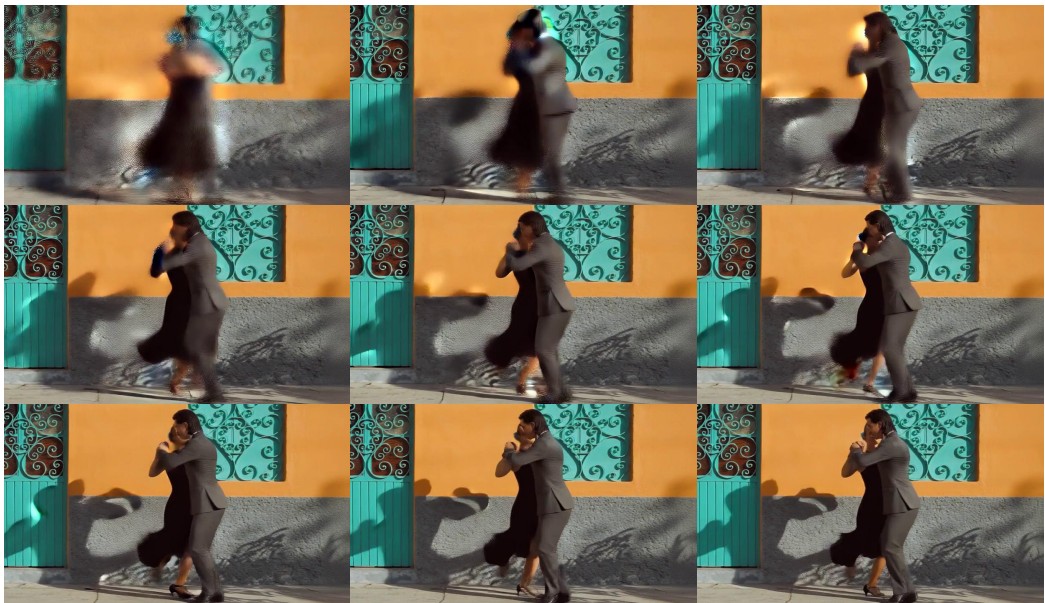

Figure 11: Qualitative visualization of Wan2.1-I2V-14B-480P across different denoising timesteps. The model captures the global layout in early stages and progressively refines fine details in later stages. Around $t/T = 40\%$, it begins to predict nearly accurate velocity fields, producing clearer frames with increasing attention to detail.

prompt specification. The validated images are subsequently converted into short dance sequences using Wan2.1-I2V-14B-480P, yielding synthetic yet realistic multi-human videos.

**Clip segmentation**. The collected videos are segmented into fixed-length clips of 49 frames. This process is guided by both the consistency of human counts and the confidence scores from the pose detector. Concretely, we traverse frames sequentially; as long as the detected number of humans remains unchanged and the pose detector reports high confidence, we continue accumulating frames. Once either criterion fails, the current segment ends and a new one begins from the following frame.

**Quality-based filtering**. To further ensure high-quality data, each clip is evaluated under four complementary metrics: (1) Aesthetic score predicted by the LAION-Aesthetics model (Schuhmann et al., 2022), a CLIP-based linear estimator widely used to assess visual appeal; (2) Optical flow magnitude computed with UniMatch (Xu et al., 2023), which measures the degree of motion and filters out overly static clips; (3) Laplacian blur score obtained via the Laplacian operator in OpenCV, which detects blurry frames and eliminates videos with poor sharpness; and (4) OCR text ratio predicted by CRAFT (Baek et al., 2019), which quantifies the proportion of textual regions and removes clips dominated by overlayed text. The thresholds for these four metrics are set to 4.5, 2.5, 50, and 0.1, respectively. Any clip failing to meet any threshold is discarded.

**Manual validation**. To safeguard the reliability of the automated pipeline, we further conduct human inspection. Specifically, we randomly sample 200 clips from the automatically curated pool and carefully examine them to verify both the visual quality and the accuracy of the pose detector. This step provides additional assurance that the dataset maintains high fidelity across multiple dimensions.

Through this pipeline, we ultimately construct a dataset comprising 46 hours of multi-human video, spanning a diverse range of interaction patterns and scenes (e.g., boxing, fencing, and dancing), while rigorously adhering to strict quality standards.

### I.2 DUALDYNAMICS BENCHMARK

Our benchmark is exclusively composed of humanoid characters, specifically designed to evaluate a model's capability in understanding motion dynamics and its robustness under diverse and chal-

lenging conditions. To this end, we collaborated with a professional animation team to curate a high-quality benchmark. From their productions, we manually selected 300 video clips, each with a fixed length of 49 frames.

The benchmark encompasses a wide spectrum of humanoid forms and artistic styles, including pixelated stick figures, anthropomorphic animals, mechanical robots, anime characters, and 3D animated avatars. These characters are placed in richly interactive scenarios that involve frequent occlusions, overlaps, and intricate inter-character coordination. Such diversity not only tests whether a model can generalize beyond standard human shapes, but also examines its ability to preserve motion control and maintain character identity under heavy interaction.

By covering varied visual domains and challenging motion patterns, the benchmark provides a rigorous testbed for assessing both the fidelity and robustness of multi-humanoid animation models.

## J    STRATEGIES FOR MEMORY- AND COMPUTE-EFFICIENT TRAINING

To alleviate the memory and computational bottlenecks in training, we adopt several strategies. First, since the T5 text encoder consumes a large amount of GPU memory, we pre-encode a set of generic prompts (e.g., "multiple humans performing interactive actions") into text embeddings before training. During training, we randomly sample from these pre-computed embeddings and inject them into the DiT model, which avoids repeatedly invoking T5 and thus significantly reduces memory usage while improving training efficiency. In addition, we enable gradient checkpointing to further lower memory overhead by trading off computation for reduced activation storage. When decoding video latents through the 3D VAE, although the runtime overhead is relatively small, preserving complex gradients introduces considerable memory cost; therefore, during training, we only decode the first two latents to extract the corresponding motion latents, effectively alleviating this issue. Finally, the overall training is conducted with the DeepSpeed framework using the ZeRO-2 optimization strategy, which ensures efficient memory utilization and stable scaling across multiple GPUs.

## K    LIMITATIONS

While MotionWeaver demonstrates strong performance across a wide range of scenarios, it still exhibits several limitations, as illustrated in Figure 12.

First, our method tends to generate blurry hands in real-world human scenes. This limitation arises from two main factors: (1) the Wan2.1-I2V-14B-480P base model itself exhibits weak generation capability for hands, and (2) our method does not explicitly control hand movements in real-world human scenarios. Unlike RealisDance (Zhou et al., 2024), which leverages HaMeR to provide detailed hand-level control signals, MotionWeaver does not incorporate such information during training. This design choice is deliberate: our primary focus is on multi-humanoid scenarios, where the hand structures of humanoid characters can differ significantly from those of humans. In such cases, enforcing hand-specific supervision would introduce a strong human-centric bias and ultimately compromise generalization to diverse humanoid forms. Second, similar to most diffusion-based frameworks, MotionWeaver is computationally expensive and incurs non-trivial inference latency. This constraint poses challenges for real-time or interactive applications, although it remains acceptable for offline content creation.

## L    BROADER IMPACTS

MotionWeaver enables pose-controllable video generation of multiple humanoid characters by modeling in 4D space and learning identity-agnostic 4D motion representations. This capability unlocks new possibilities across a wide spectrum of applications, including creative content production, professional animation, immersive gaming, virtual reality/augmented reality experiences, digital avatars for social interaction, virtual cinematography, and human–robot interaction simulation. In parallel, we introduce DualDynamics, the first dedicated benchmark for multi-humanoid image animation. DualDynamics is deliberately designed to assess a model's generalization ability at inference time,

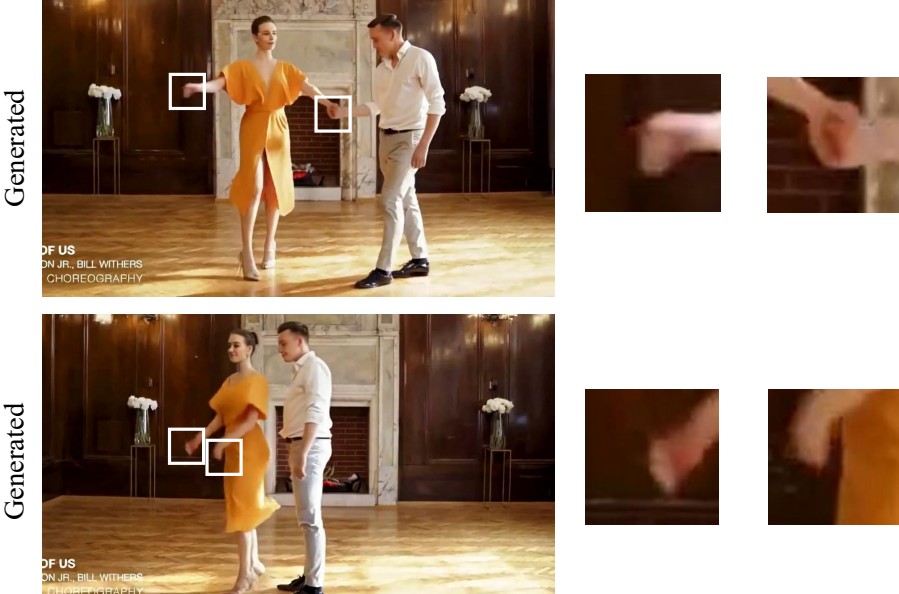

Figure 12: MotionWeaver tends to produce blurry hands in real-world human scenes due to weak base-model hand generation and the absence of explicit hand-level control, a deliberate choice to avoid human-centric bias and preserve generalization across diverse humanoid forms.

thereby serving as a valuable resource for advancing research and benchmarking future methods in this emerging field.

Nevertheless, our model is also capable of producing realistic multi-human videos, which raises potential concerns about misuse. These risks include, but are not limited to, deepfake creation, identity impersonation, and privacy infringements. To mitigate such risks, we plan to release public model checkpoints with visible watermarks, and to accompany the release of code and models with clear ethical guidelines and explicit usage licenses. Under these safeguards, we believe MotionWeaver can serve as a responsible tool that not only pushes forward scientific research but also fosters innovation in the creative industries.

## M  USER STUDY

To gain an accurate understanding of user preferences, we conducted a comprehensive survey assessing five key perceptual dimensions: Visual Quality, Motion Alignment, Interaction Coherence, Occlusion Handling, and Character Preservation. We compared our proposed framework, MotionWeaver, against two competitive baselines, RealisDance-DiT and UniAnimate-DiT. The results, shown in Figure 13, illustrate the proportion of total votes received by each method. Across all five evaluation dimensions, MotionWeaver consistently secures the largest share of user preference. These findings underscore the strength of our framework in balancing both identity preservation and motion fidelity, thereby demonstrating its effectiveness in aligning with human perceptual expectations for multi-humanoid animation.

## N  MORE DETAILS OF DYNAMIC C-ROPE

The HSI module is designed to establish a unified 4D space between video latents and motion tokens. It consists of two complementary components: (1) a depth-aware attention mechanism that implicitly encodes the z-axis (depth), and (2) Dynamic C-RoPE, which explicitly models the remaining temporal and spatial axes $(t, x, y)$.

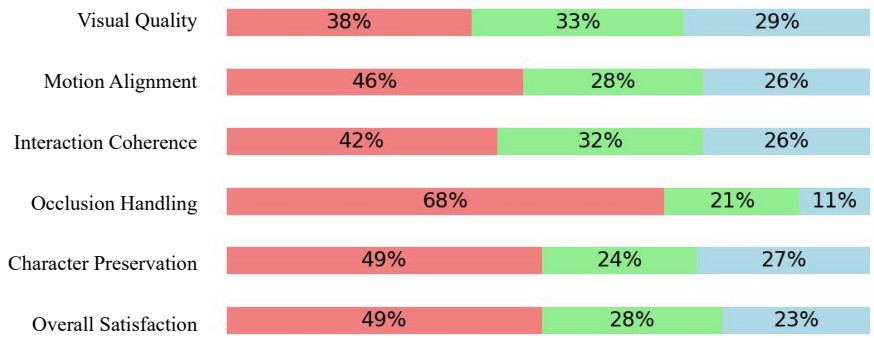

Figure 13: User preferences across visual quality, motion alignment, interaction coherence, occlusion handling, and character preservation for different methods.

Unlike conventional RoPE, which primarily encodes relative positional relationships in self-attention layers, Dynamic C-RoPE is incorporated into cross-attention to explicitly capture positional dependencies between queries and keys across different modalities. Inspired by Wan2.1-I2V-14B-480P, we divide each query and key embedding into three subspaces within the multi-head cross-attention mechanism. Each subspace is rotated using the corresponding rotary matrices, enabling simultaneous encoding of temporal and spatial relations.

A critical challenge in applying RoPE to cross-attention is the alignment of queries and keys within a shared coordinate system. Specifically, query positions $(x, y)$ are naturally defined in the image coordinate system, whereas key positions $(x, y)$ are defined in the camera coordinate system. To bridge this discrepancy, we leverage the depth value $z$ of each key to project its $(x, y)$ coordinates onto the image plane. Consequently, identical $(t, x, y)$ values from queries and keys correspond to the same spatiotemporal location, ensuring consistent positional alignment across modalities.

In our setting, queries with the same index in the attention computation are associated with a fixed rotary matrix, making them static. By contrast, keys with the same index may map to different positions due to varying motion dynamics, rendering them inherently dynamic. To account for this discrepancy, we dynamically select the appropriate rotary matrix for each key based on its positional information before applying the rotation. This ensures that the encoded representations accurately reflect the relative spatiotemporal structure.

To further clarify this process, we provide code that illustrates how the Dynamic C-RoPE procedure applies dynamic rotation to keys:

Listing 1: Illustration of Dynamic C-RoPE procedure: dynamic rotation of keys

```python
def Dynamic_C_RoPE_key(x, grid_sizes, freqs, motion_pos):
    b, n, c = x.size(0), x.size(2), x.size(3) // 2
    ps = motion_pos.size(1)
    ps_token_len = x.size(1)
    token_len = ps_token_len // ps

    freqs = freqs.split([c - 2 * (c // 3), c // 3, c // 3], dim=1)
    output = []

    for i, (f, h, w) in enumerate(grid_sizes.tolist()):
        ## Rotary matrix candidate pool
        freqs_i_fxy = torch.cat([
            freqs[0][:f].view(f, 1, 1, -1).expand(f, h, w, -1),
            freqs[1][:h].view(1, h, 1, -1).expand(f, h, w, -1),
            freqs[2][:w].view(1, 1, w, -1).expand(f, h, w, -1)
            ],
                dim=-1).reshape(f, h, w, 1, -1)
```

```
        x_i = torch.view_as_complex(x[i].to(torch.float32
            ).reshape(ps, token_len, n, -1, 2))
        m_i = motion_pos[i]
        scales = torch.tensor([w-1, h-1, 0], device=m_i.device
            ).view(1, 1, 3)
        m_i_scaled = (m_i * scales).long()
        t_idx = torch.arange(f, device=m_i.device
            ).view(1, f).expand(ps, f)
        h_idx = m_i_scaled[..., 1]
        w_idx = m_i_scaled[..., 0]
        ## dynamiclly select
        freqs_fxy_selected = freqs_i_fxy[t_idx, h_idx, w_idx]
        d1,d2,d3,d4 = freqs_fxy_selected.shape
        freqs_selected = freqs_fxy_selected.unsqueeze(2
            ).repeat(1,1,24,1,1).reshape(d1,d2*24,d3,d4)

        ## rotate the keys
        x_i = torch.view_as_real(x_i*freqs_selected).flatten(3)
        d1,d2,d3,d4=x_i.shape
        x_i = x_i.reshape(d1*d2,d3,d4)
        output.append(x_i)
    return torch.stack(output).float()
```

## O  QUANTITATIVE EVALUATION METRICS

The detailed formulations of the evaluation metrics are presented below.

- **FVD↓**:
  We evaluate temporal realism using features from the Inflated 3D ConvNet (I3D). Each video is partitioned into non-overlapping 16-frame segments, and embeddings are extracted. The empirical means $\mu_r, \mu_f$ and covariances $\Sigma_r, \Sigma_f$ are computed for real and generated sets, and the distance is

$$\text{FVD} = \|\mu_r - \mu_f\|_2^2 + \text{Tr}\big(\Sigma_r + \Sigma_f - 2(\Sigma_r \Sigma_f)^{1/2}\big). \tag{11}$$

- **FVD-VID↓**:
  Instead of treating short clips independently, we first average all I3D embeddings belonging to the same video, resulting in one descriptor per sequence. Statistics $\mu_r, \mu_f, \Sigma_r, \Sigma_f$ are then estimated across the video-level features, and the Fréchet distance is calculated as

$$\text{FVD}-\text{VID} = \|\mu_r - \mu_f\|_2^2 + \text{Tr}\big(\Sigma_r + \Sigma_f - 2(\Sigma_r \Sigma_f)^{1/2}\big). \tag{12}$$

- **FID↓**:
  For each individual frame, we extract activations from a pretrained Inception-V3 network. By aggregating these embeddings into Gaussian distributions $(\mu_r, \Sigma_r)$ and $(\mu_f, \Sigma_f)$, we define

$$\text{FID} = \|\mu_r - \mu_f\|_2^2 + \text{Tr}\big(\Sigma_r + \Sigma_f - 2(\Sigma_r \Sigma_f)^{1/2}\big). \tag{13}$$

- **L1↓**:
  Given a ground-truth video $I_t$ and a generated sequence $\hat{I}_t$, both of size $T \times H \times W \times C$, the pixel-wise reconstruction error is

$$\text{L1} = \frac{1}{255 \times THWC} \sum_{t=1}^{T} \sum_{x=1}^{W} \sum_{y=1}^{H} \sum_{c=1}^{C} \big|I_t(x,y,c) - \hat{I}_t(x,y,c)\big|. \tag{14}$$

- **PSNR↑**:
  We first compute the mean squared error for each frame:

$$\text{MSE} = \frac{1}{HWC} \sum_{x=1}^{W} \sum_{y=1}^{H} \sum_{c=1}^{C} \big(I_t(x,y,c) - \hat{I}_t(x,y,c)\big)^2. \tag{15}$$

Peak signal-to-noise ratio is then defined as

$$\text{PSNR} = 20 \log_{10}\left(\frac{255}{\sqrt{\text{MSE}}}\right). \tag{16}$$

- **SSIM↑**:
  For each frame $t$ and color channel $c$, we measure structural similarity:

$$\text{SSIM}_t^c = \text{SSIM}\big(I_t(\cdot, \cdot, c),\ \hat{I}_t(\cdot, \cdot, c)\big). \tag{17}$$

  The reported score is the average across all frames and channels:

$$\text{SSIM} = \frac{1}{TC}\sum_{t=1}^{T}\sum_{c=1}^{C}\text{SSIM}_t^c. \tag{18}$$

- **LPIPS↓**:
  For each video frame, let $\phi_\ell(\cdot)$ be the feature map from the $\ell$-th layer of a pretrained backbone, and let $w_\ell$ denote the learned channel-wise weights. The frame-level perceptual distance is defined as

$$l_t = \frac{1}{L}\sum_{\ell=1}^{L}\frac{1}{H_\ell W_\ell}\big\| w_\ell \odot \big(\phi_\ell(I_t) - \phi_\ell(\hat{I}_t)\big)\big\|_1. \tag{19}$$

  The video-level LPIPS score is then obtained by averaging across all frames:

$$\text{LPIPS} = \frac{1}{T}\sum_{t=1}^{T}l_t. \tag{20}$$

- **CLIP↑**:
  At each frame $t$, both the ground-truth and generated images are embedded with the CLIP image encoder, yielding $v_t, \hat{v}_t \in \mathbb{R}^d$. After $\ell_2$ normalization, the similarity is measured by cosine distance:

$$s_t = \frac{v_t^\top \hat{v}_t}{\|v_t\|\,\|\hat{v}_t\|}. \tag{21}$$

  The final CLIP score for the video is computed as the temporal average:

$$\text{CLIP} = \frac{1}{T}\sum_{t=1}^{T}s_t. \tag{22}$$

## P  MORE QUALITATIVE RESULTS

We present detailed qualitative comparisons on the DualDynamics benchmark dataset in Figures 14, 15, and 16. Additional qualitative results on out-of-benchmark data are shown in Figures 17, 18, 19, and 20. Furthermore, extended results of MotionWeaver on the benchmark dataset are provided in Figures 21, and 22, while more qualitative inference results on out-of-benchmark data are also included in Figure 23. For frame-by-frame comparisons, we refer readers to the supplementary videos.

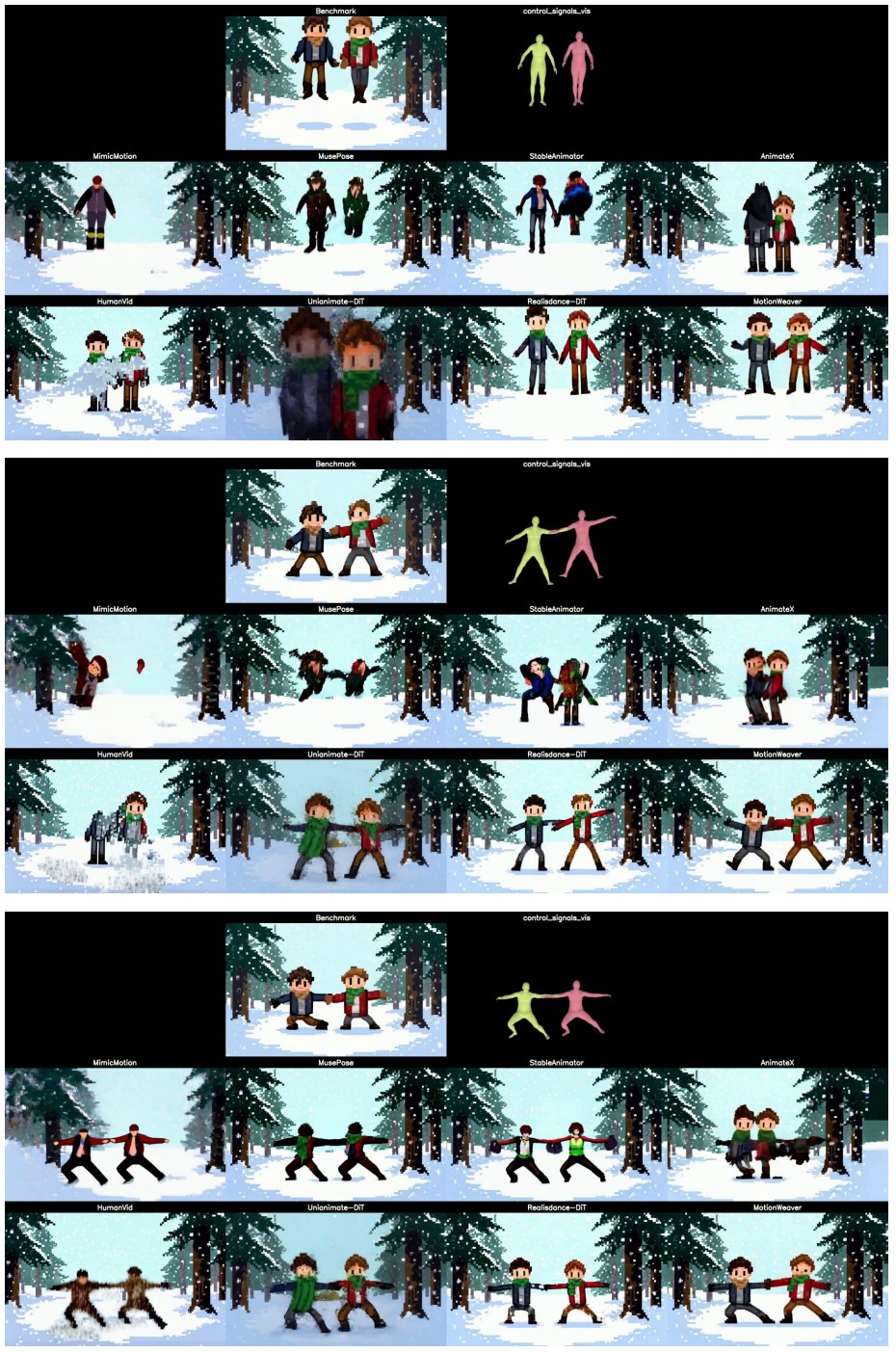

Figure 14: Qualitative comparisons on the DualDynamics benchmark dataset. From top to bottom and left to right: benchmark video (ground truth), target motion, MimicMotion, MusePose, StableAnimator, Animate-X, HumanVid, UniAnimate-DiT, RealisDance-DiT, and our MotionWeaver.

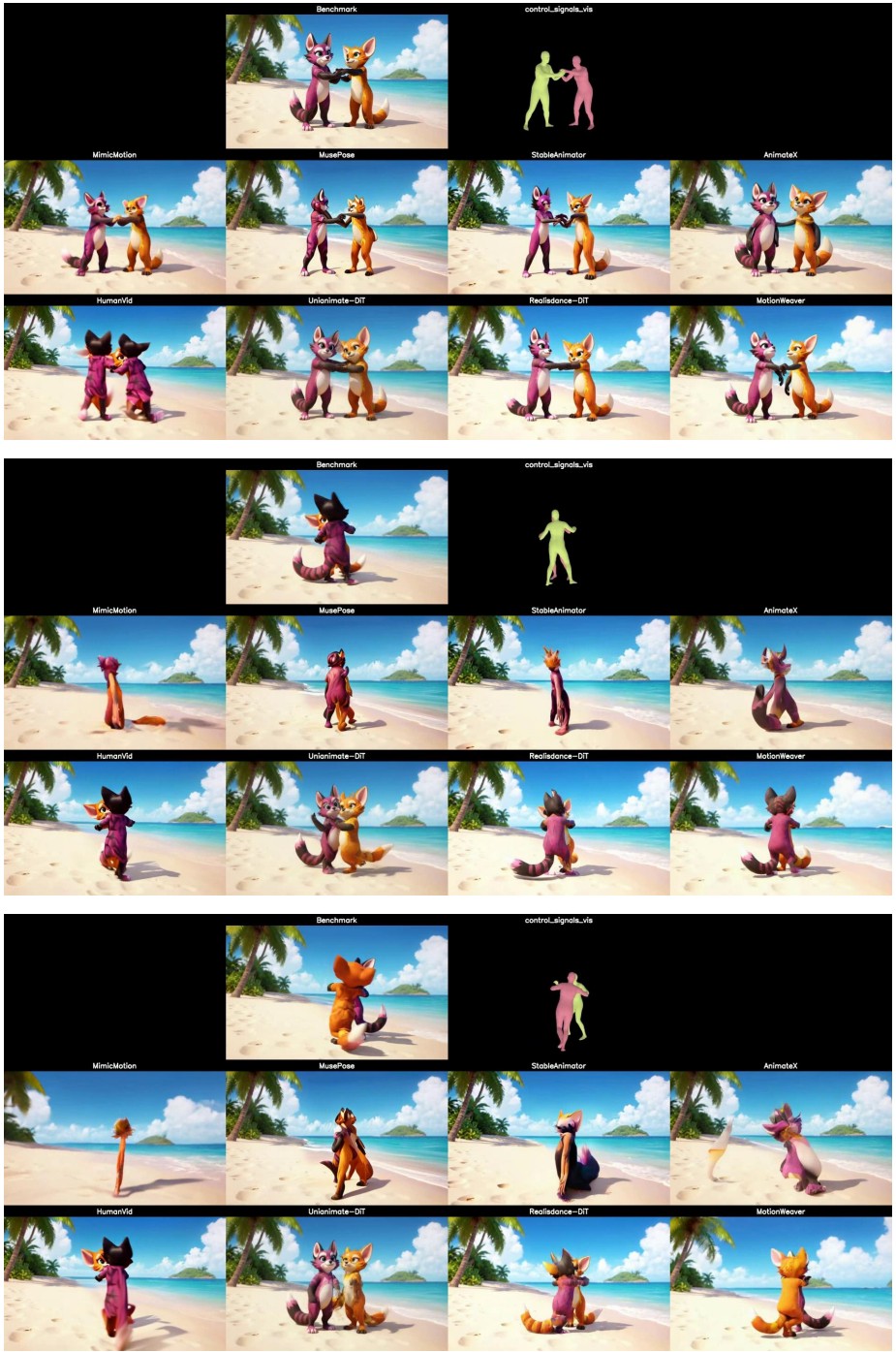

Figure 15: Qualitative comparisons on the DualDynamics benchmark dataset. From top to bottom and left to right: benchmark video (ground truth), target motion, MimicMotion, MusePose, StableAnimator, Animate-X, HumanVid, UniAnimate-DiT, RealisDance-DiT, and our MotionWeaver.

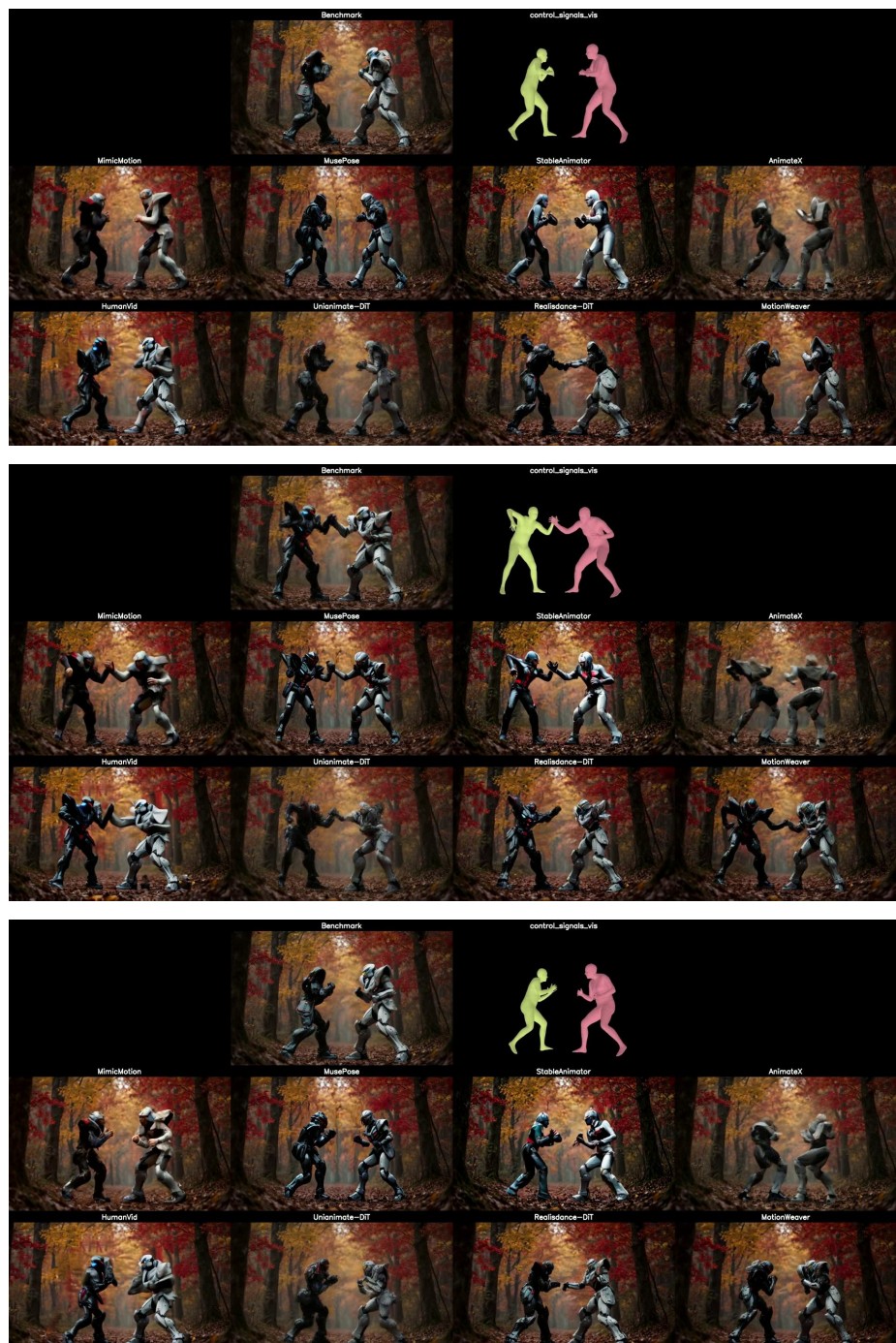

Figure 16: Qualitative comparisons on the DualDynamics benchmark dataset. From top to bottom and left to right: benchmark video (ground truth), target motion, MimicMotion, MusePose, StableAnimator, Animate-X, HumanVid, UniAnimate-DiT, RealisDance-DiT, and our MotionWeaver.

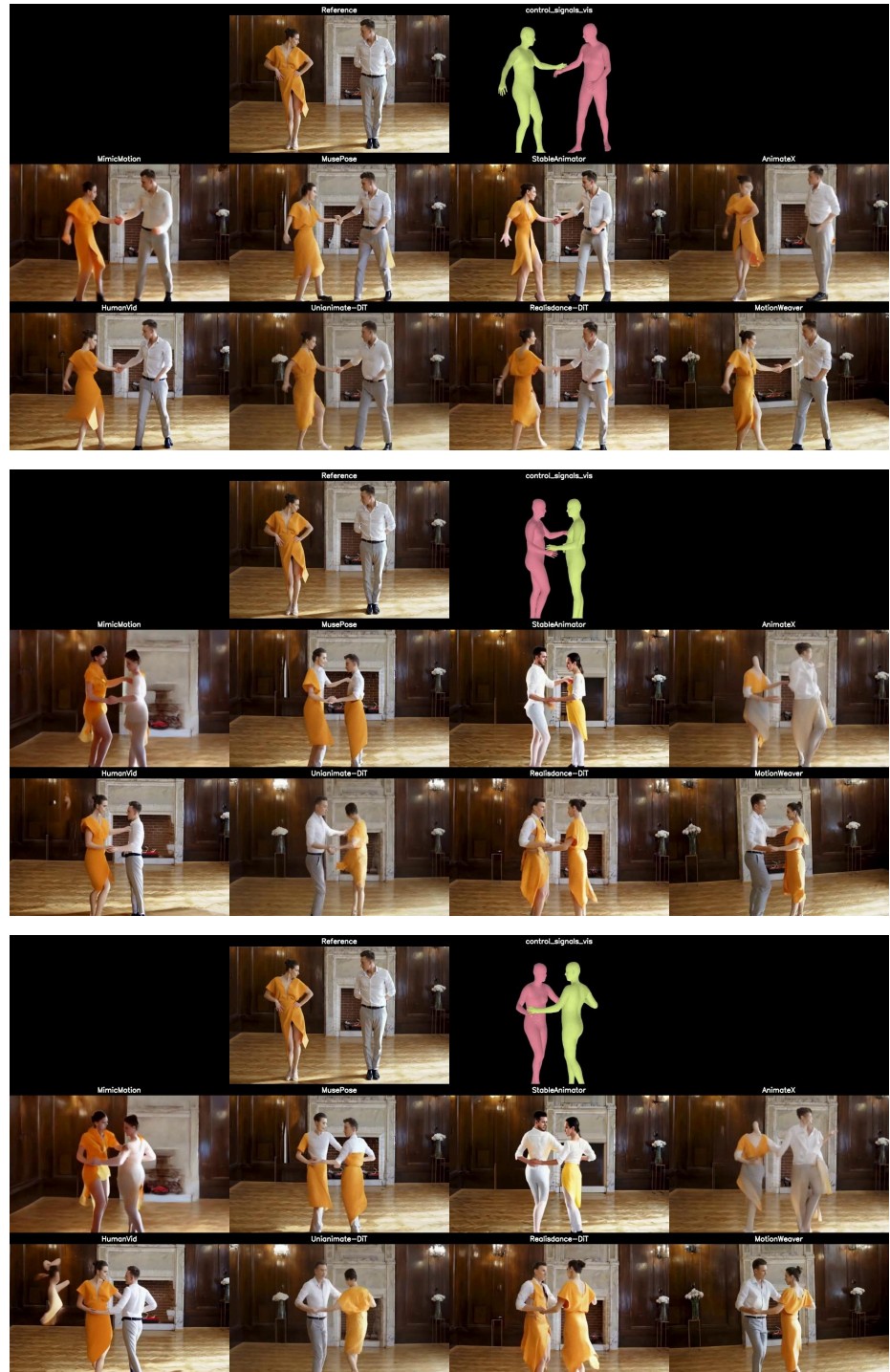

Figure 17: Qualitative results. From top to bottom and left to right: reference image, target motion, MimicMotion, MusePose, StableAnimator, AnimateX, HumanVid, UniAnimate-DiT, RealisDance-DiT, and our MotionWeaver.

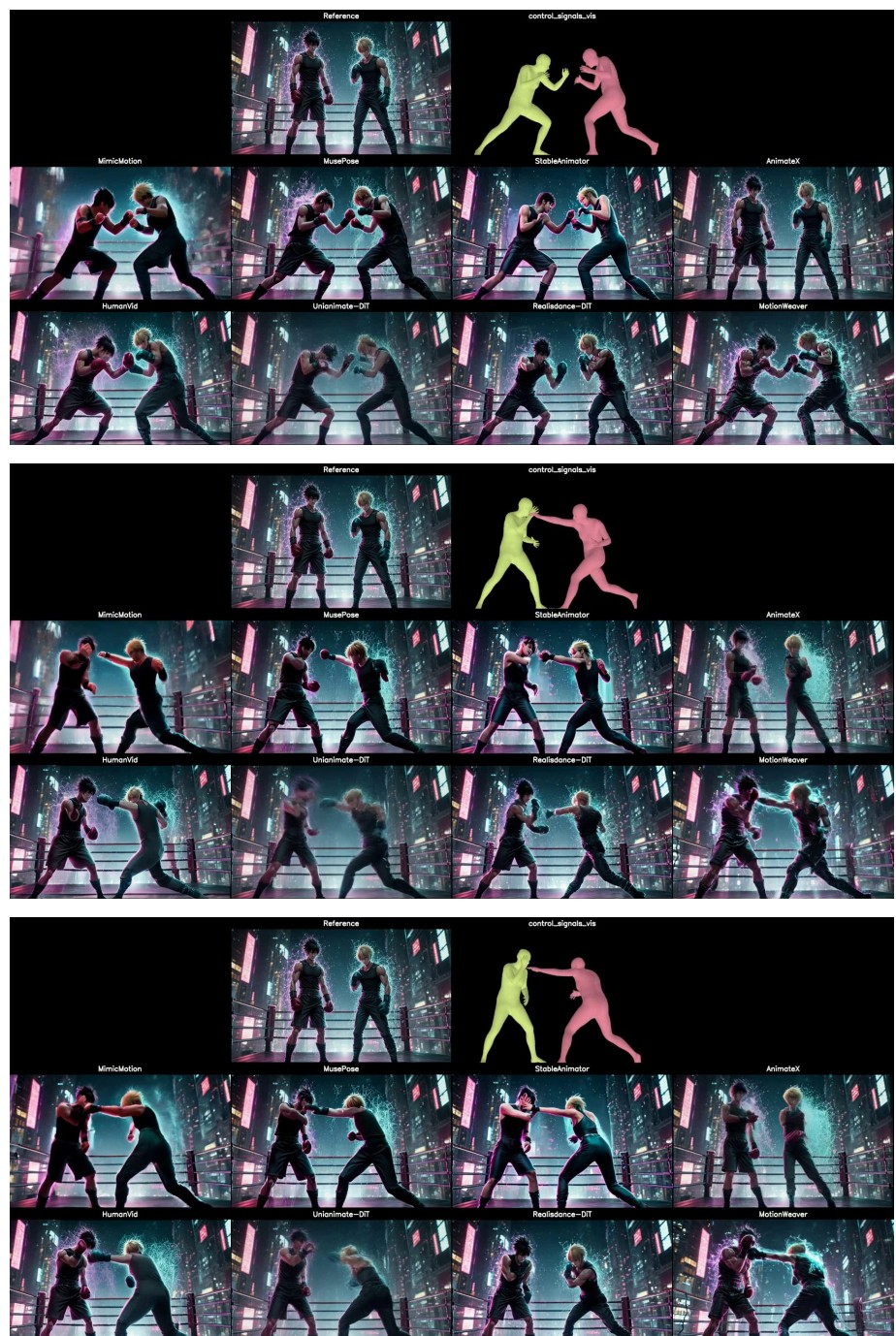

Figure 18: Qualitative results. From top to bottom and left to right: reference image, target motion, MimicMotion, MusePose, StableAnimator, AnimateX, HumanVid, UniAnimate-DiT, RealisDance-DiT, and our MotionWeaver.

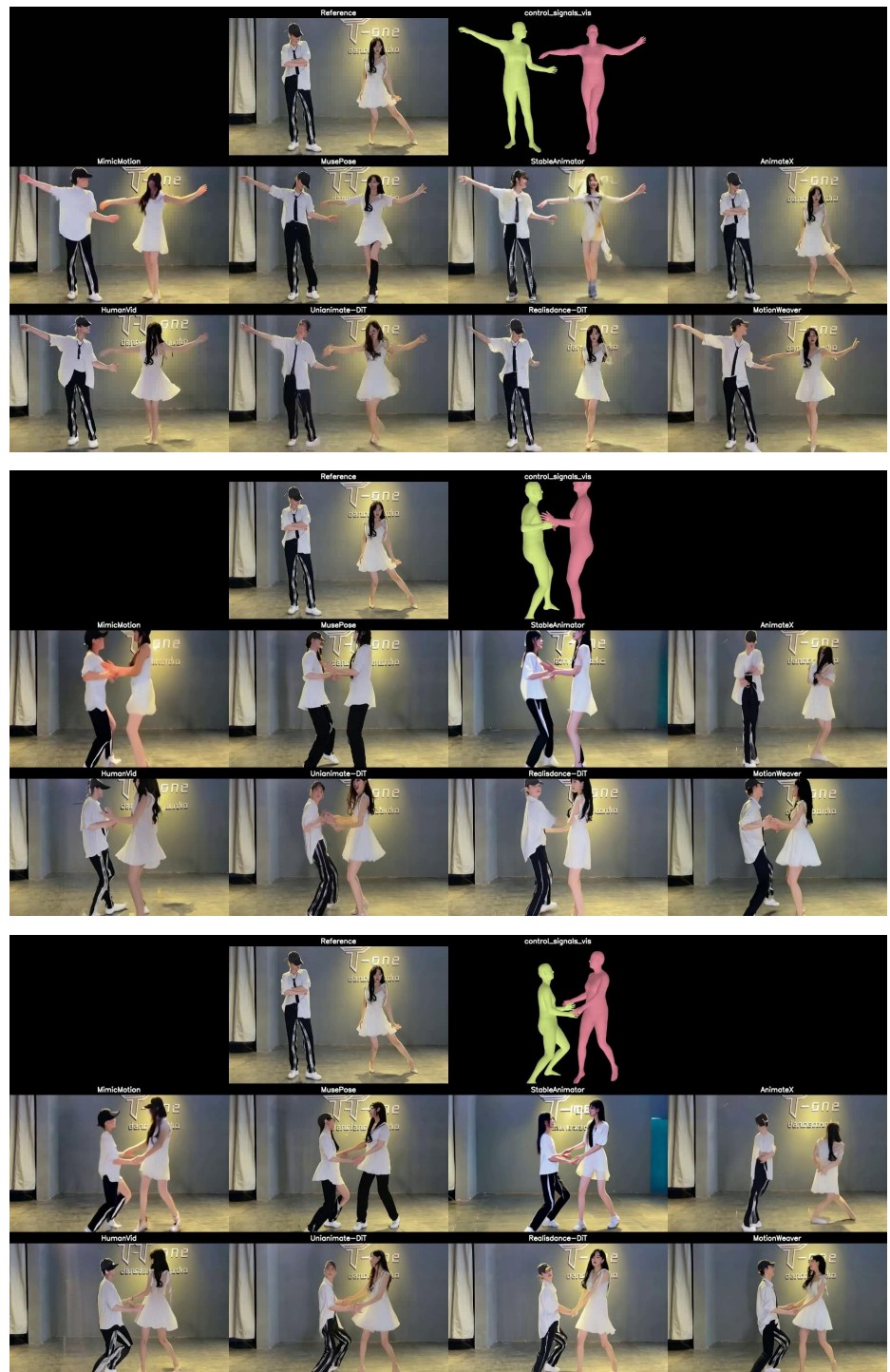

Figure 19: Qualitative results. From top to bottom and left to right: reference image, target motion, MimicMotion, MusePose, StableAnimator, AnimateX, HumanVid, UniAnimate-DiT, RealisDance-DiT, and our MotionWeaver.

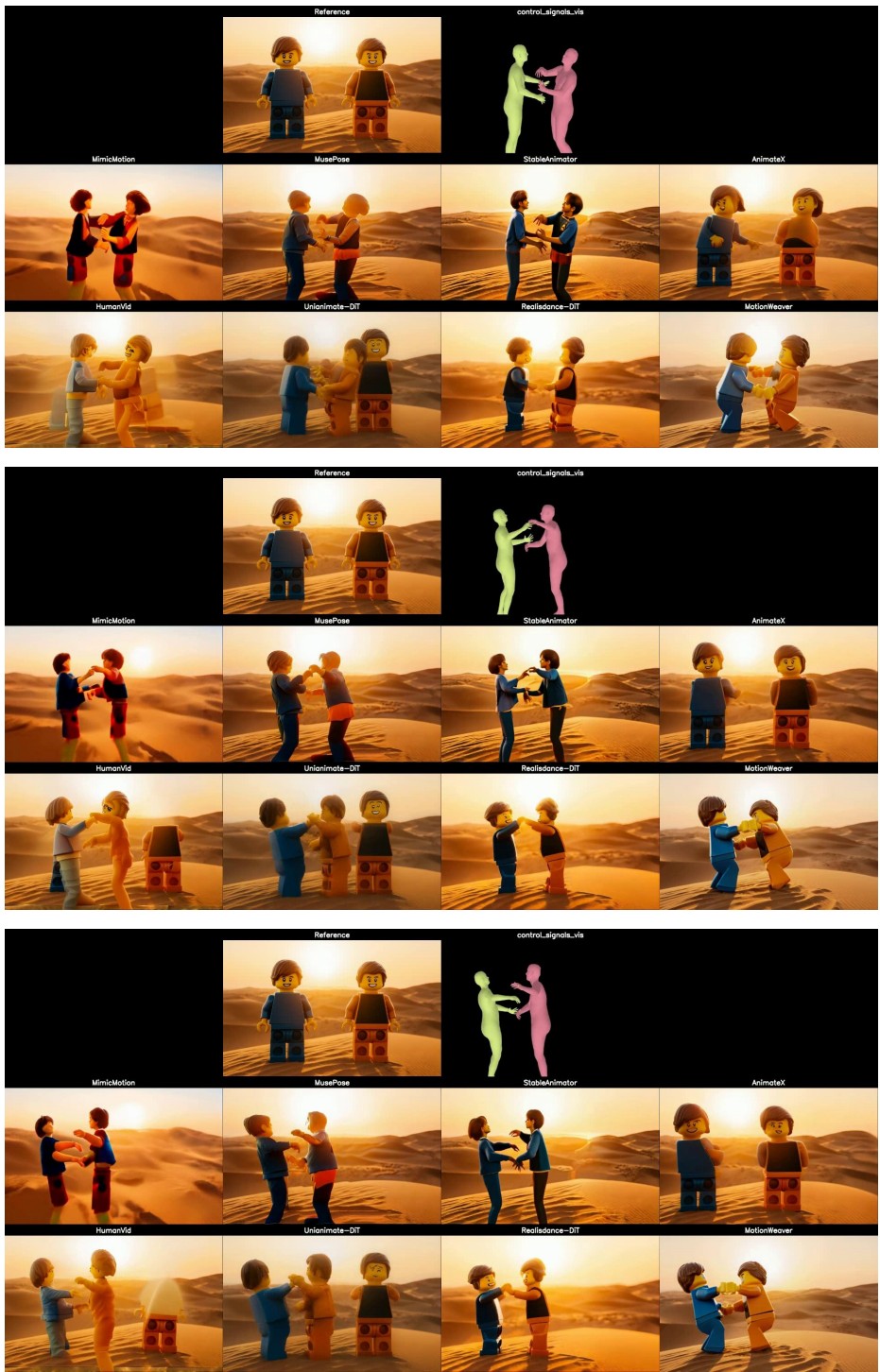

Figure 20: Qualitative results. From top to bottom and left to right: reference image, target motion, MimicMotion, MusePose, StableAnimator, AnimateX, HumanVid, UniAnimate-DiT, RealisDance-DiT, and our MotionWeaver.

Benchmark     Motion     Generated

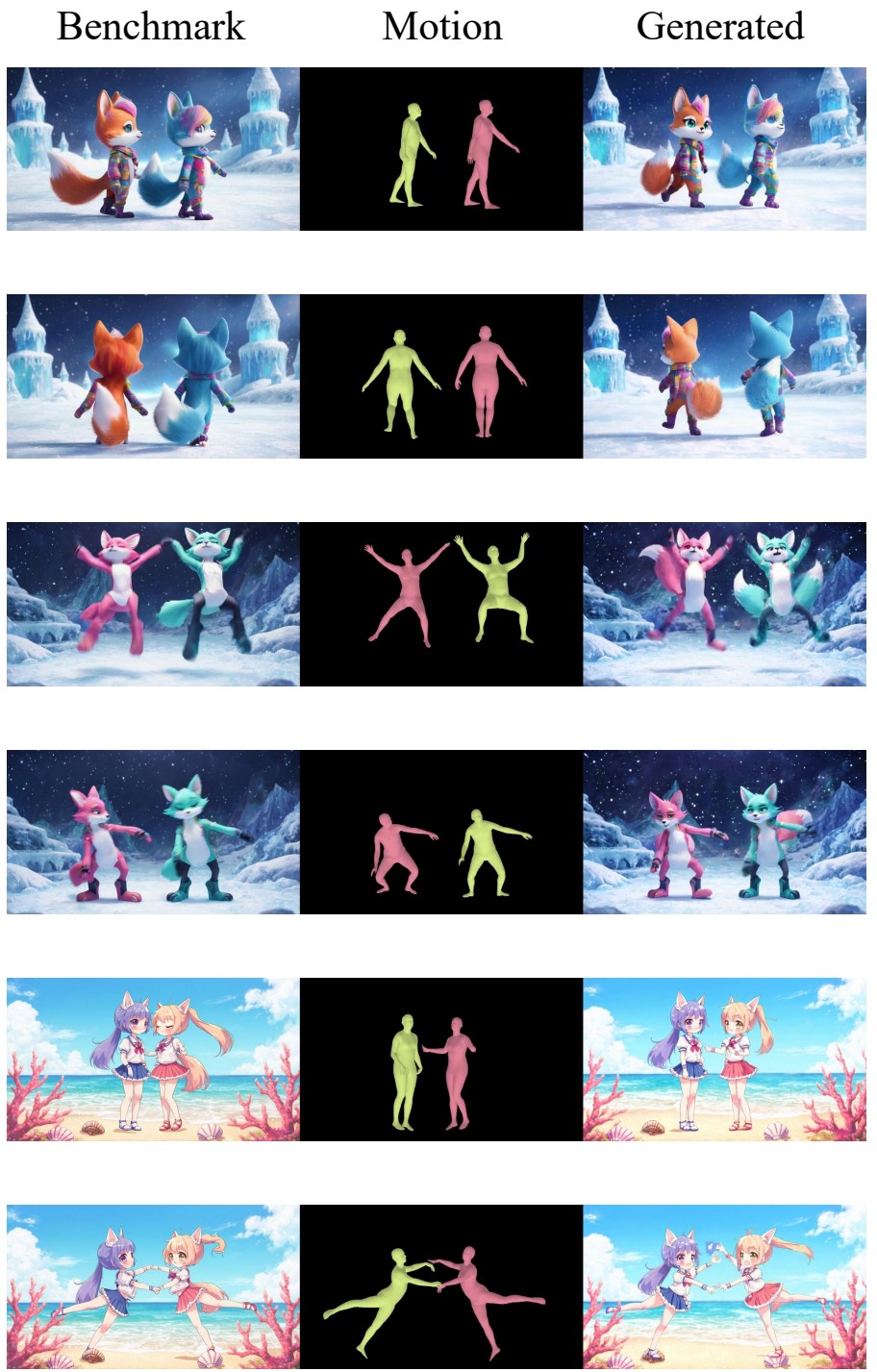

Figure 21: Qualitative results of MotionWeaver on the DualDynamics benchmark dataset.

Benchmark    Motion    Generated

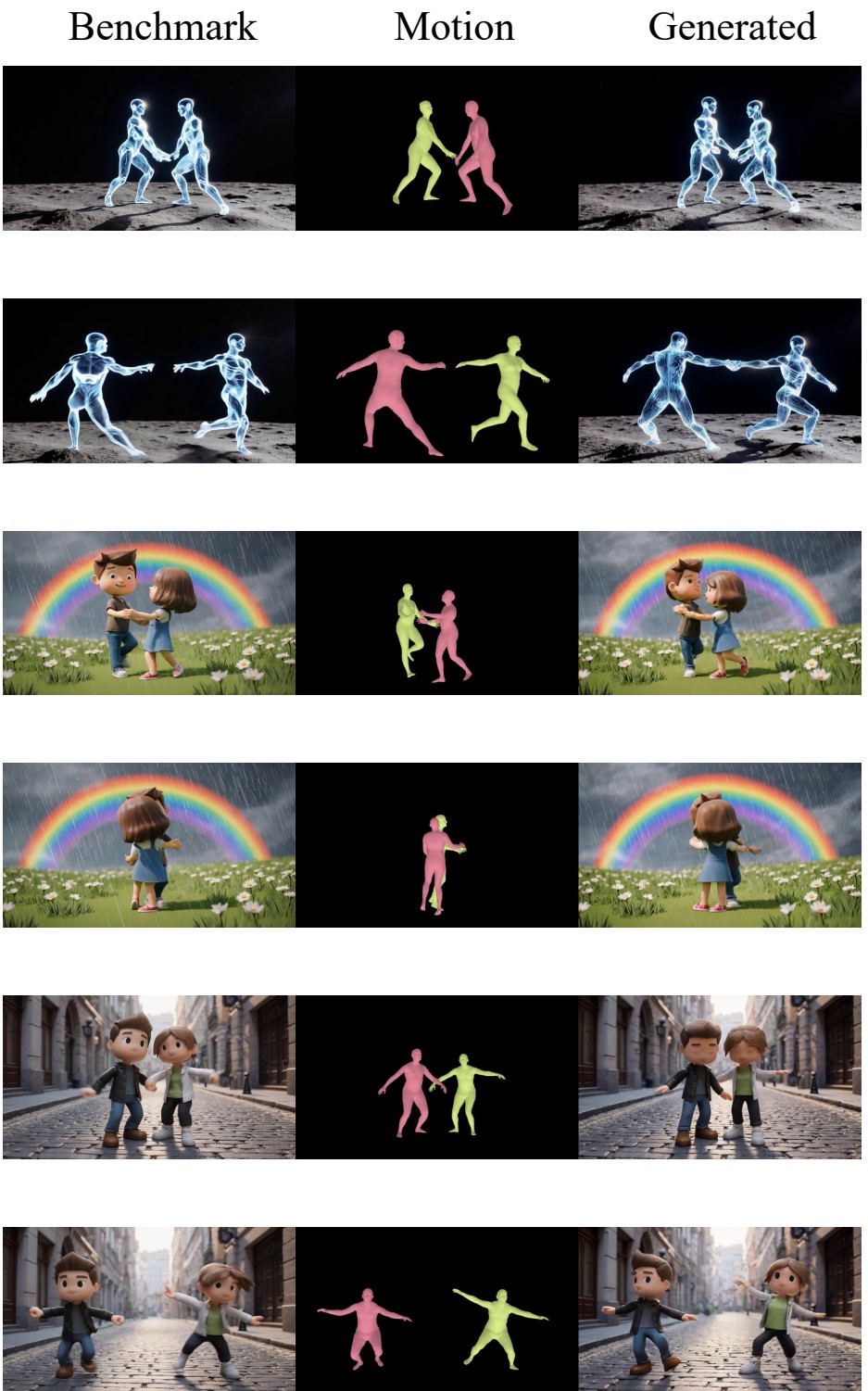

Figure 22: Qualitative results of MotionWeaver on the DualDynamics benchmark dataset.

| Reference Image | Motion | Generated | Motion | Generated |
| --- | --- | --- | --- | --- |

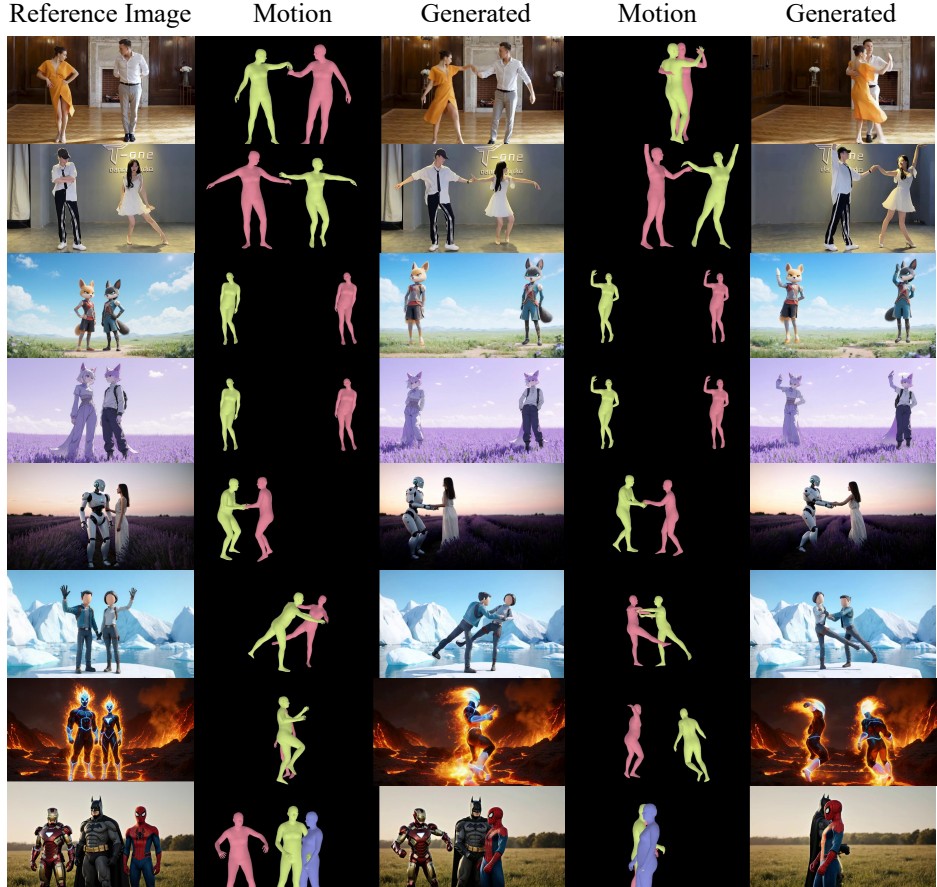

Figure 23: Qualitative results of MotionWeaver.

## Q  SCENE DESCRIPTIONS FOR IMAGE GENERATION PROMPTS

The scene descriptions used in the prompts for image generation are provided here, offering a concise overview of the input conditions. For more details on how these prompts are utilized in our framework, please refer to Section I.1.

- in a dense tropical rainforest
- on the snowy peaks of the Himalayas
- in a canyon carved by a river
- beside a tranquil bamboo grove
- on a frozen lake under the moonlight
- in a valley filled with morning mist
- by a hot spring in the mountains
- on the shoreline with crashing waves
- in a meadow full of fireflies at night
- on a winding mountain trail
- in a jungle with ancient ruins
- on a golden desert dune at sunset
- beside a river with drifting lotus leaves
- in a field of colorful tulips
- on cliffs facing the endless sea
- in a forest with red and golden autumn leaves
- by a glacier with floating icebergs
- in a canyon glowing with sunset light
- on a hillside covered with tea plantations
- in a peach orchard during full bloom
- by a secluded waterfall hidden in the jungle
- in a pine forest after snowfall
- on a plateau with yaks grazing
- beside a calm fjord surrounded by mountains
- in a sunflower field stretching to the horizon
- on a volcanic island with black sand beaches
- in a lavender field under the evening sky
- in a cherry blossom park during spring
- beside a crystal-clear stream with pebbles
- in a meadow with a rainbow after rain
- on a prairie with galloping wild horses
- in a canyon with a turquoise river
- on a coral beach with seashells
- under giant sequoia trees
- in a field covered with morning dew
- floating on clouds above the world
- inside an ancient library filled with scrolls
- in a desert oasis with palm trees
- on a bridge made of glowing light

- in a realm of floating islands
- inside a crystal ice cave
- in a volcanic landscape with flowing lava
- on the surface of the moon
- in a garden of giant mushrooms
- inside a glowing underwater city
- inside a traditional Japanese teahouse
- in a medieval European castle courtyard
- on a quiet cobblestone street
- on a rooftop overlooking skyscrapers
- in a cozy countryside farmhouse
- at a festive lantern festival
- in a futuristic cyberpunk alley
- inside a grand opera house

