# OpenReview forum: "MotionWeaver: Holistic 4D-Anchored Framework for Multi-Humanoid Image Animation"
_ICLR.cc/2026/Conference — ICLR 2026 Poster_

### Official Review · Reviewer_dGPY · 2025-10-28

**Soundness:** 3
**Presentation:** 3
**Contribution:** 3
**Rating:** 4
**Confidence:** 4

**Summary:**

This paper introduces Motion Weaver, a novel and holistic 4D-Anchored Framework designed for multi-humanoid image animation. It specifically tackles the challenges of inter-character occlusion, complex interactions, and diverse humanoid forms—issues that often cause existing single-person animation models to fail. The method utilizes a Unified Choreography Core (UCC), a Hyper-Scene Integrator (HSI), and a Hierarchical 4D Supervision (H4S) strategy, alongside contributing the MultiHuman46 dataset and DualDynamics benchmark. The results are qualitatively impressive in multi-character settings.

**Strengths:**

1.	The paper tackles the crucial and challenging problem of multi-character image animation, which current state-of-the-art methods typically cannot handle robustly due to their focus on single-person scenarios.
2.	The introduction of the MultiHuman46 dataset and the DualDynamics benchmark provides essential resources for advancing research in the multi-humanoid animation field.

**Weaknesses:**

1.	The proposed Motion Weaver is trained on the dedicated, multi-humanoid focused MultiHuman46 dataset. The methods chosen for comparison (e.g., RealisDance-DiT, UniAnimate-DiT, etc.) are predominantly designed and trained on single-person or general video datasets. Comparing Motion Weaver (trained on multi-person data) against baselines (trained on single-person data) on a multi-person benchmark (DualDynamics) creates a massive advantage for the proposed method. The baseline models are being tested significantly Out-of-Distribution (OOD) for the task of multi-person interaction and occlusion handling. The reported performance gain is likely inflated due to this training data mismatch.
2.	The authors must perform a fair comparison by either: a) Re-training/Fine-tuning all comparable baselines on the MultiHuman46 training set and reporting the results. This would isolate the performance difference due to the architectural design. b) Alternatively, the authors should present an ablation study where Motion Weaver is only trained on a standard, single-person dataset and then tested on the multi-person benchmark, to demonstrate the robustness of its 4D framework even without specific multi-person training data.
3.	The paper focuses almost exclusively on multi-humanoid scenarios. It is critical to demonstrate that the novel 4D-anchored components and mechanisms designed for multiple characters do not degrade performance or introduce unnecessary complexity/overhead when applied to standard, single-person animation tasks.

**Questions:**

Please refer to Weaknesses for more details. If my concerns are solved, I am glad to raise my score.

---

> ### Author Response · Authors · 2025-11-20
> **Response to Weakness 1 and Weakness 2**
>
> We are sincerely grateful for the valuable time and effort you devoted to improving the quality of this paper.
>
> We have thoroughly conducted the additional experiments you suggested and ensured every concern is fully addressed. We hope our responses meet your satisfaction.
>
> > Weakness: Unfair comparison due to training data mismatch
>
> Thank you for your valuable comment and insightful suggestions !
>
>
>
>
>
> ## *Quick Summary of the Response*
>
> To address this concern, we have **strictly followed your suggestion** and performed additional experiments. Our response is summarized in three key points:
>
> 1. **Fair Comparison via Fine-tuning:** We fine-tuned RealisDance-DiT and UniAnimate-DiT on our MultiHuman46 dataset. Both quantitative and qualitative results confirm that even with domain-specific training, these methods still **struggle with complex multi-humanoid challenges** and consistently underperform MotionWeaver.
> 2. **Theoretical Analysis:** We provide an in-depth **analysis of method design** to theoretically explain why existing baselines are fundamentally limited in addressing this task and substantiate the inherent advantages of our 4D framework.
> 3. **Holistic System Contribution:** We emphasize that our contribution extends far beyond a simple model design improvement. We present a comprehensive system that **encompasses** a novel task definition, theoretical foundations, specialized data collection, and model design.
>
>
>
>
>
> ## *Fair Comparison via Fine-tuning*
>
> Given the substantial time and computational resources required to fine-tune all baseline models, we selected the two strongest state-of-the-art methods, UniAnimate-DiT and RealisDance-DiT, for this comparative analysis. These models were subsequently **fine-tuned on our specialized multi-human dataset**, MultiHuman46.
>
> **As demonstrated in (Fig. 8, page 19)**, these baselines remain **unable** to robustly handle the core challenges inherent in multi-humanoid animation, including diverse humanoid forms, complex interactions, and frequent occlusions, even after being trained on task-specific data. As shown in Table 1, while the performance metrics of the baselines were elevated after fine-tuning, **they remain comprehensively weaker than MotionWeaver**.
>
>
>
> **Table 1**: Quantitative results in DualDynamics benchmark.
>
> |                    Model                     |   FVD ↓   | FID-VID ↓ |   FID ↓   |    L1 ↓    |  PSNR ↑   |   SSIM ↑   |  LPIPS ↓   |
> | :------------------------------------------: | :-------: | :-------: | :-------: | :--------: | :-------: | :--------: | :--------: |
> | $\text{UniAnimate-DiT}_{\text{fine-tuned}}$  |   162.9   |   22.11   |   22.47   |   0.5314   |   29.15   |   0.5412   |   0.3277   |
> | $\text{RealisDance-DiT}_{\text{fine-tuned}}$ |   157.1   |   21.16   |   21.83   |   0.5213   |   29.11   |   0.5401   |   0.3263   |
> |            $\text{MotionWeaver}$             | **145.7** | **20.34** | **19.41** | **0.4836** | **29.19** | **0.5428** | **0.3213** |

---

> ### Author Response · Authors · 2025-11-20
> **Response to Weakness 1 and Weakness 2**
>
> ## *Theoretical Analysis*
>
> Previous methods are inherently designed for single-human animation, making them **fundamentally deficient** in supporting complex multi-humanoid tasks.
>
> Taking UniAnimate-DiT as a representative example, it places multiple skeletons onto a single image plane and simply concatenates the encoded features with video latents, a design that fundamentally fails to account for the complexities of multi-humanoid scenarios, leading to critical limitations:
>
> 1. Lacking an explicit binding mechanism, the model struggles to maintain motion-character consistency during position swaps.
> 2. Without explicit depth modeling, the model fails to resolve occlusion ordering (front/back confusion) and generates artifacts, when skeletons overlap.
> 3. Reliance on 2D control images and 2D pixel-wise MSE loss lacks explicit 4D supervision and inadvertently couples motion with appearance, impairing true motion understanding.
>
> In contrast, our holistic 4D-anchored framework incorporates novel and essential designs to address these gaps:
>
> 1. Explicitly binds motion tokens to their corresponding characters via group attention (UCC).
> 2. Constructs a shared 4D space to intricately fuse motion tokens with video latents, modeling the precise 4D position of each motion-unit (HSI).
> 3. Leverages 4D motion tokens as control signals (UCC), and provides explicit supervision at both the 4D and motion levels (H4S).
>
> **These novel and essential designs are the key to effectively tackle multi-humanoid animation challenges**.
>
>
>
>
>
> ## *Holistic System Contribution*
>
> Our work delivers a comprehensive method, not merely an isolated model architectural improvement. Therefore, the comparison can be regarded as reflecting the strengths of this **complete system**, rather than narrowly focusing on model design alone. Specifically, our end-to-end system includes contributions across multiple aspects:
>
> 1. Task Introduction: We are the first to propose this **novel and significant task**: multi-humanoid animation involving diverse humanoid forms, complex interactions, and frequent occlusions.
> 2. Problem Definition and Analysis: We define this new problem and provide an in-depth analysis of its challenges, as detailed in our Introduction.
> 3. Data and Benchmark: We designed a novel and systematic pipeline for the automated collection, filtering, and processing of multi-human interaction data, alongside a standard benchmark. This comprehensive pipeline and benchmark are fully detailed in the Appendix to facilitate future research.
> 4. Model Design: Our model incorporates several necessary components specifically to address the multi-humanoid challenge, which single-human methods fundamentally lack.

---

> ### Author Response · Authors · 2025-11-20
> **Response to Weakness 3**
>
> > Weakness 3: *It is critical to demonstrate that the novel 4D-anchored components designed for multiple characters do not degrade performance or introduce unnecessary complexity/overhead when applied to standard, single-person animation tasks.*
>
> Thank you for raising this important point!
>
> ## *Performance*
>
> We conducted both qualitative (Fig.10 ,page 20) and quantitative (Tables 1 and 2) experiments on the standard single-person TikTok and Fashion benchmarks. Given the long duration of minimal motion in the Fashion dataset videos, we first created a refined test set by extracting segments with significant human action changes.
>
> Our approach achieves competitive performance in standard single-person animation tasks, unequivocally **demonstrating no performance degradation** compared to state-of-the-art specialized methods.
>
> **Table 1**: **Single-human setting (TikTok)**, a scenario that prior methods were explicitly tailored for.
>
> |      Model      |   FVD ↓    |   FID ↓   |  PSNR ↑   |  SSIM ↑   |  LPIPS ↓  |
> | :-------------: | :--------: | :-------: | :-------: | :-------: | :-------: |
> |   MimicMotion   |   472.51   |   34.88   |   19.30   |   0.751   |   0.220   |
> |    MusePose     |   532.75   |   53.50   |   18.20   |   0.757   |   0.248   |
> |    Animate-X    |   508.63   |   32.77   |   16.71   |   0.743   |   0.285   |
> | UniAnimate-DiT  |   402.14   | **28.47** | **19.35** |   0.765   |   0.235   |
> | RealisDance-DiT |   458.81   |   34.39   |   17.55   |   0.717   |   0.261   |
> |  MotionWeaver   | **391.25** |   28.49   |   19.11   | **0.782** | **0.214** |
>
> **Table 2: Single-human setting (Fashion)**, a scenario that prior methods were explicitly tailored for.
>
> |      Model      |   FVD ↓    |   FID ↓   |  PSNR ↑   |  SSIM ↑   |  LPIPS ↓  |
> | :-------------: | :--------: | :-------: | :-------: | :-------: | :-------: |
> |   MimicMotion   |   481.05   |   50.11   |   17.92   |   0.866   |   0.106   |
> |    MusePose     |   475.88   |   34.13   |   17.75   |   0.849   |   0.083   |
> |    Animate-X    |   544.33   |   42.28   |   17.75   |   0.842   |   0.085   |
> | UniAnimate-DiT  |   579.20   |   68.03   | 18.54 |   0.665   |   0.377   |
> | RealisDance-DiT |   439.96   | **34.05** |   **18.65**   |   0.844   |   0.127   |
> |  MotionWeaver   | **429.34** |   34.45   |   18.21   | **0.867** | **0.082** |
>
>
>
> ## *Complexity/overhead*
>
> While our design in the training phase is sophisticated, the **inference pipeline remains straightforward and light**, primarily consisting of three stages:
>
> - Extracts motion tokens.
>
> - Binds motion tokens and their corresponding characters using **group attention module**.
>
> - Fuses motion tokens and video latents.
>
> *As you can see, **only** the group attention module is specifically introduced for multi-character support.*
>
> **Complexity**: The group attention module is simply composed of a single cross-attention layer followed by an MLP.
>
> **Overhead**: Compared to the heavy and multi-round denoising process, this component is extremely lightweight and computed only once. Its calculation time accounts for **less than 1\%** of the total inference duration, making the overhead **negligible** for standard single-person animation tasks.

---

> > ### Comment · Reviewer_dGPY · 2025-11-26
> >
> > Thank the authors for the response. My concerns are solved, I am glad to raise my score to 6.

---

> > > ### Author Response · Authors · 2025-11-26
> > >
> > > Dear Reviewer dGPY,
> > >
> > > Thank you very much! We sincerely appreciate your recognition of the value of our work and the constructive feedback you provided throughout this process. Your thoughtful insights have not only helped improve this manuscript but also inspired us to further advance this research direction. We are committed to continuing to refine this method and look forward to contributing more impactful work in the future.
> > >
> > > Best regards,
> > >
> > > MotionWeaver Authors

---

### Official Review · Reviewer_9egx · 2025-10-28

**Soundness:** 3
**Presentation:** 3
**Contribution:** 3
**Rating:** 6
**Confidence:** 4

**Summary:**

The paper proposes MotionWeaver for multi-humanoid image animation. MotionWeaver introduces unified motion representations to extract identity-agnostic 4D motions and bind them to corresponding characters, enabling generalization across diverse humanoid forms. It fuses motion representations with video latents to control the generated motion. The paper also introduces training datasets and benchmarks to support this task. Results show the effectiveness of the proposed method.

**Strengths:**

1. The multi-humanoid image animation is an important, practical and interesting task.
2. The proposed method designs several components to extract motion to improve generalization, which is well motivated and reasonable.
3. The results show that the proposed method outperforms previous methods.

**Weaknesses:**

1. The training setting of comparing methods seems to be different from the proposed method. Are the comparing methods trained on the same data?
2. The proposed method consists several steps. What is the inference time comparison to previous methods? How would the error propagation affect the final results?
3. The evaluation is only conducted on the proposed benchmark. It is unclear whether the proposed method outperforms comparing methods on existing benchmarks.

**Questions:**

1. What is the training settings of the comparing methods? Are they trained on the same data? Could the improvements come from different training data?
2. What is the inference time comparison to previous methods? How would the error propagation affect the final results?
3. What is performance of the proposed method on existing benchmarks?

---

> ### Author Response · Authors · 2025-11-20
> **Response to Weakness 1 and Question 1 (1/2)**
>
> We deeply appreciate your dedicated time and effort in providing such insightful feedback. **Your comments have been crucial in solidifying the contribution and presentation of our work**. We assure you that all revised discussions and newly presented results will be integrated into the final paper.
>
> > Weakness 1 and Question 1: *What is the training settings of the comparing methods? Are they trained on the same data? Could the improvements come from different training data?*
>
> Thank you for these insightful questions.
>
> ## *Quick Summary of the Response*
>
> We acknowledge that some of the improvement is attributable to the training data. **To address your reasonable concern**, our response is summarized in three key points:
>
> 1. **Fair Comparison via Fine-tuning:** We fine-tuned RealisDance-DiT and UniAnimate-DiT on our MultiHuman46 dataset. Both quantitative and qualitative results confirm that even with domain-specific training, these methods still **struggle with complex multi-humanoid challenges** and consistently underperform MotionWeaver.
> 2. **Theoretical Analysis:** We provide an in-depth **analysis of method design** to theoretically explain why existing baselines are fundamentally limited in addressing this task and substantiate the inherent advantages of our 4D framework.
> 3. **Holistic System Contribution:** We emphasize that our contribution extends far beyond a simple model design improvement. We present a comprehensive system that **encompasses** a novel task definition, theoretical foundations, specialized data collection, and model design.
>
>
>
>
>
> ## *Fair Comparison via Fine-tuning*
>
> Given the substantial time and computational resources required to fine-tune all baseline models, we selected the two strongest state-of-the-art methods, **UniAnimate-DiT** and **RealisDance-DiT**, for this comparative analysis. These models were subsequently **fine-tuned on our specialized multi-human dataset**, MultiHuman46.
>
> **As demonstrated in （Fig. 8, page 19)**, these baselines remain **unable** to robustly handle the core challenges inherent in multi-humanoid animation, including diverse humanoid forms, complex interactions, and frequent occlusions, even after being trained on task-specific data. As shown in Table 1, while the performance metrics of the baselines were elevated after fine-tuning, **they remain comprehensively weaker than MotionWeaver**.
>
>
>
> **Table 1**: Quantitative results in DualDynamics benchmark.
>
> |                    Model                     |   FVD ↓   | FID-VID ↓ |   FID ↓   |    L1 ↓    |  PSNR ↑   |   SSIM ↑   |  LPIPS ↓   |
> | :------------------------------------------: | :-------: | :-------: | :-------: | :--------: | :-------: | :--------: | :--------: |
> | $\text{UniAnimate-DiT}_{\text{fine-tuned}}$  |   162.9   |   22.11   |   22.47   |   0.5314   |   29.15   |   0.5412   |   0.3277   |
> | $\text{RealisDance-DiT}_{\text{fine-tuned}}$ |   157.1   |   21.16   |   21.83   |   0.5213   |   29.11   |   0.5401   |   0.3263   |
> |            $\text{MotionWeaver}$             | **145.7** | **20.34** | **19.41** | **0.4836** | **29.19** | **0.5428** | **0.3213** |

---

> ### Author Response · Authors · 2025-11-20
> **Response to Weakness 1 and Question 1 (2/2)**
>
> *Following the discussion above*
>
> ## *Theoretical Analysis*
>
> Previous methods are inherently designed for single-human animation, making them **fundamentally deficient** in supporting complex multi-humanoid tasks.
>
> Taking UniAnimate-DiT as a representative example, it places multiple skeletons onto a single image plane and simply concatenates the encoded features with video latents, a design that fundamentally fails to account for the complexities of multi-humanoid scenarios, leading to critical limitations:
>
> 1. Lacking an explicit binding mechanism, the model struggles to maintain motion-character consistency during position swaps.
> 2. Without explicit depth modeling, the model fails to resolve occlusion ordering (front/back confusion) and generates artifacts, when skeletons overlap.
> 3. Reliance on 2D control images and 2D pixel-wise MSE loss lacks explicit 4D supervision and inadvertently couples motion with appearance, impairing true motion understanding.
>
> In contrast, our holistic 4D-anchored framework incorporates **novel and essential** designs to address these gaps:
>
> 1. Explicitly binds motion tokens to their corresponding characters via group attention (UCC).
> 2. Constructs a shared 4D space to intricately fuse motion tokens with video latents, modeling the precise 4D position of each motion-unit (HSI).
> 3. Leverages 4D motion tokens as control signals (UCC), and provides explicit supervision at both the 4D and motion levels (H4S).
>
> **These novel and essential designs are the key to effectively tackle multi-humanoid animation challenges**.
>
>
>
>
>
>
>
> ## *Holistic System Contribution*
>
> Our work delivers a comprehensive method, not merely an isolated model architectural improvement. Therefore, the comparison can be regarded as reflecting the strengths of this **complete system**, rather than narrowly focusing on model design alone. Specifically, our end-to-end system includes contributions across multiple aspects:
>
> 1. Task Introduction: We are the first to propose this **novel and significant task**: multi-humanoid animation involving diverse humanoid forms, complex interactions, and frequent occlusions.
> 2. Problem Definition and Analysis: We define this new problem and provide an in-depth analysis of its challenges, as detailed in our Introduction.
> 3. Data and Benchmark: We designed a novel and systematic pipeline for the automated collection, filtering, and processing of multi-human interaction data, alongside a standard benchmark. This comprehensive pipeline and benchmark are fully detailed in the Appendix to facilitate future research.
> 4. Model Design: Our model incorporates several necessary components specifically to address the multi-humanoid challenge, which single-human methods fundamentally lack.

---

> > ### Author Response · Authors · 2025-11-20
> > **Response to Weakness 2 and Question 2**
> >
> > > (a) Weakness 2 and Question 2:  *The proposed method consists several steps. What is the inference time comparison to previous methods?*
> > >
> > > (b) *How would the error propagation affect the final results?*
> >
> > Thank you for your questions.
> >
> >
> >
> >
> >
> > ## *Regarding Q (a): Inference Time*
> >
> > While our method's training phase is complex and sophisticated, the overall inference process is **straightforward and symmetrical to previous methods**. Specifically, the steps involve
> >
> > 1. UCC extracting unified motion representations
> > 2. HSI integrating these to guide generation.
> >
> > We have provided inference time comparisons with prior work in Table 2. Note that UniAnimate-DiT, RealisDance-DiT, and MotionWeaver are all based on the same foundational model, Wan2.1-I2V-14B. All experiments were rigorously tested on the NVIDIA H100 GPU. However, the differences in measured time are primarily attributed to variations in specific **implementation** details, such as TeaCache and VRAM management.
> >
> > **Table 2**: Comparison of inference time.
> >
> > |     Method     | UniAnimate-DiT | RealisDance-DiT | MotionWeaver |
> > | :------------: | :------------: | :-------------: | :----------: |
> > | Inference time |    3min 49s    |    2min 26s     |   2min 41s   |
> >
> >
> >
> >
> >
> > ## *Regarding Q (b): Error Propagation*
> >
> > Since our inference process is simple and straightforward, the primary source of potential error lies with the external pose detector.
> >
> > Crucially, the UCC's **motion tokenizer (based on VQVAE)** effectively mitigates this issue. By performing temporal downsampling and quantization, the tokenizer inherently leverages temporal coherence, allowing it to suppress isolated, single-frame detection errors using information from neighboring frames. This mechanism robustly inhibits the propagation of upstream errors.
> >
> > We have provided empirical validation in (Fig. 9, page 19). By intentionally corrupting the detection result of a single frame to simulate an occasional detector error, we demonstrated that the final generated video remained coherent and retained accurate pose alignment, confirming the framework's resilience to external noise.

---

> ### Author Response · Authors · 2025-11-20
> **Response to Weakness 3 and Question 3**
>
> > Weakness 3 and Question 3: *What is performance of the proposed method on existing benchmarks?*
>
> Thank you for your valuable comment.
>
> The existing public benchmarks in human image animation are **limited to** **TikTok** and **Fashion**, both of which are primarily designed to evaluate **single-human setting**.
>
> **Following your insightful suggestions**, we have conducted additional qualitative (Fig. 10, page 20) and quantitative (Table 1 and 2) experiments on the Fashion and TikTok datasets. Given the long duration of minimal motion in the Fashion dataset videos, we first created a refined test set by extracting segments with significant human action changes.
>
> Our method is **primarily designed for and demonstrably superior in the challenging multi-humanoid setting**. However, these new evaluations further substantiate that our approach is also capable of achieving competitive results in the single-human setting, which has been the focus of prior work.
>
> **Table 1**: **Single-human setting (TikTok)**, a scenario that prior methods were explicitly tailored for.
>
> |      Model      |   FVD ↓    |   FID ↓   |  PSNR ↑   |  SSIM ↑   |  LPIPS ↓  |
> | :-------------: | :--------: | :-------: | :-------: | :-------: | :-------: |
> |   MimicMotion   |   472.51   |   34.88   |   19.30   |   0.751   |   0.220   |
> |    MusePose     |   532.75   |   53.50   |   18.20   |   0.757   |   0.248   |
> |    Animate-X    |   508.63   |   32.77   |   16.71   |   0.743   |   0.285   |
> | UniAnimate-DiT  |   402.14   | **28.47** | **19.35** |   0.765   |   0.235   |
> | RealisDance-DiT |   458.81   |   34.39   |   17.55   |   0.717   |   0.261   |
> |  MotionWeaver   | **391.25** |   28.49   |   19.11   | **0.782** | **0.214** |
>
> **Table 2: Single-human setting (Fashion)**, a scenario that prior methods were explicitly tailored for.
>
> |      Model      |   FVD ↓    |   FID ↓   |  PSNR ↑   |  SSIM ↑   |  LPIPS ↓  |
> | :-------------: | :--------: | :-------: | :-------: | :-------: | :-------: |
> |   MimicMotion   |   481.05   |   50.11   |   17.92   |   0.866   |   0.106   |
> |    MusePose     |   475.88   |   34.13   |   17.75   |   0.849   |   0.083   |
> |    Animate-X    |   544.33   |   42.28   |   17.75   |   0.842   |   0.085   |
> | UniAnimate-DiT  |   579.20   |   68.03   | 18.54 |   0.665   |   0.377   |
> | RealisDance-DiT |   439.96   | **34.05** |   **18.65**   |   0.844   |   0.127   |
> |  MotionWeaver   | **429.34** |   34.45   |   18.21   | **0.867** | **0.082** |

---

> > ### Comment · Reviewer_9egx · 2025-11-21
> >
> > Thanks for the rebuttal. The authors' response provided new results and comparisons, which addressed my main concerns. And I update my rating.

---

> > > ### Author Response · Authors · 2025-11-22
> > >
> > > Thanks a lot for the suggestions and valuable comments! We are pleased to know that our responses have addressed your questions. We appreciate for your decision to raise the score!

---

### Official Review · Reviewer_FnGY · 2025-10-29

**Soundness:** 3
**Presentation:** 3
**Contribution:** 3
**Rating:** 8
**Confidence:** 4

**Summary:**

The paper presented MotionWeaver which synthesizes multi-character video given the reference character images and pose sequences. The proposed MotionWeaver includes these modules:

1)	Unified-Choreopgraphy Core (UCC), which extracts 4D motion representations of characters that exclude their appearances. UCC uses SMPL to represent 3D human body as their joint coordinates and normalizes to a standardized skeleton. Individual character is segmented from the reference image to bind with the motion representations by a pre-trained I2V model.

2)	Hyper-Scene Integrator (HIS), which estimates the depth from the occlusion-loss supervision and encodes with a dynamic C-RoPE. This representation allows a Hierachical-4DSupervision (H4S) to train the model with occlusion loss and motion-level loss, etc.

The MotionWeaver model is trained on a new 46-hour multi-human video dataset and evaluated on a new DualDynamics benchmark including 300 videos with complicated interactions between two humanoid characters. MotionWeaver outperforms 7 recent character animation models on this new DualDynamics benchmark.

**Strengths:**

This work is well motivated. Character video generation including two characters remains very challenging. The paper presented a clear analysis of the issues of motion representation and 4D modeling.

The proposed technical approaches, i.e., UCC, HIS, H4S, are technical novel and make sense to improve the performance of multi-character video generation.

The experiments on the new DualDynamics benchmark show clear improvement, which outperform recent related works substantially.

**Weaknesses:**

The paper only reported the performance comparison on the new DualDynamics benchmark. Why only part of the constructed benchmark will be released? Then how do subsequent works compare with MotionWeaver?

All the technical approaches, i.e., UCC, HIS, H4S, are backward applicable to the single character case. So please show the performance of MotionWeaver on Fashion and TikTok, thus, the readers have a clear understanding of the advantages of these modules.

**Questions:**

Throughout the paper, it seems all approaches and the training/benchmark datasets focused on two-humanoid interactions. So perhaps “two humanoid image animation” is a more proper claim than “multi-humanoid image animation”?

ll.251: “and augment them with the latent timestep t”, could you elaborate how to do this?

---

> ### Author Response · Authors · 2025-11-20
> **Response to Weakness 1 and Weakness 2**
>
> Your positive feedback is a great source of encouragement as we continue to refine this paper！
>
> We sincerely thank you for your kind recognition of our work and your constructive suggestions.  We hope our responses meet your satisfaction.
>
> > Weakness 1: *Why only part of the constructed benchmark will be released?*
>
> Thank you for pointing out this potential confusion. We sincerely apologize for the misunderstanding. We want to **clarify** that our DualDynamics benchmark will be **fully released** if the paper is accepted, ensuring that all subsequent works can compare against MotionWeaver.
>
> > Weakness 2: *So please show the performance of MotionWeaver on Fashion and TikTok*
>
> **Following your insightful suggestions**, we have conducted additional qualitative (Fig. 10, page 20) and quantitative (Table 1 and 2) experiments on the Fashion and TikTok datasets.
>
> Our method is primarily designed for and **demonstrably superior in the challenging multi-humanoid setting.** However, these new evaluations further substantiate that our approach is also capable of achieving competitive results in the single-human setting, a scenario that prior methods were explicitly tailored for. Given the long duration of minimal motion in the Fashion dataset videos, we first created a refined test set by extracting segments with significant human action changes.
>
> **Table 1**: **Single-human setting (TikTok)**, a scenario that prior methods were explicitly tailored for.
>
> |      Model      |   FVD ↓    |   FID ↓   |  PSNR ↑   |  SSIM ↑   |  LPIPS ↓  |
> | :-------------: | :--------: | :-------: | :-------: | :-------: | :-------: |
> |   MimicMotion   |   472.51   |   34.88   |   19.30   |   0.751   |   0.220   |
> |    MusePose     |   532.75   |   53.50   |   18.20   |   0.757   |   0.248   |
> |    Animate-X    |   508.63   |   32.77   |   16.71   |   0.743   |   0.285   |
> | UniAnimate-DiT  |   402.14   | **28.47** | **19.35** |   0.765   |   0.235   |
> | RealisDance-DiT |   458.81   |   34.39   |   17.55   |   0.717   |   0.261   |
> |  MotionWeaver   | **391.25** |   28.49   |   19.11   | **0.782** | **0.214** |
>
>
>
> **Table 2: Single-human setting (Fashion)**, a scenario that prior methods were explicitly tailored for.
>
> |      Model      |   FVD ↓    |   FID ↓   |  PSNR ↑   |  SSIM ↑   |  LPIPS ↓  |
> | :-------------: | :--------: | :-------: | :-------: | :-------: | :-------: |
> |   MimicMotion   |   481.05   |   50.11   |   17.92   |   0.866   |   0.106   |
> |    MusePose     |   475.88   |   34.13   |   17.75   |   0.849   |   0.083   |
> |    Animate-X    |   544.33   |   42.28   |   17.75   |   0.842   |   0.085   |
> | UniAnimate-DiT  |   579.20   |   68.03   | 18.54 |   0.665   |   0.377   |
> | RealisDance-DiT |   439.96   | **34.05** |   **18.65**   |   0.844   |   0.127   |
> |  MotionWeaver   | **429.34** |   34.45   |   18.21   | **0.867** | **0.082** |

---

> ### Author Response · Authors · 2025-11-20
> **Answer to Question 1 and Question 2**
>
> > Question 1: *So perhaps “two humanoid image animation” is a more proper claim than “multi-humanoid image animation”?*
>
> We sincerely appreciate the careful reading and valuable suggestion. We agree that the training process and the benchmark primarily focus on two-humanoid interactions. This is an accurate observation.
>
> However, our model's architecture was **intrinsically designed to support** multi-humanoid image animation. **We confirm that the model can indeed process scenes with more than two humanoids**, as shown in (Fig. 6, page 17).
>
> Given that our experimental validation and quantitative results predominantly leverage the two-humanoid setting, the term *"two humanoid image animation"* would indeed provide a more conservative and precisely supported claim. **We fully respect your judgment**. If, after considering the architectural design and the current experimental scope, you ultimately feel that *"two-humanoid"* is the most appropriate claim, we will **gladly adopt your suggestion** and revise the terminology throughout the paper to ensure maximum clarity and accuracy.
>
> > Question 2: *ll.251: “and augment them with the latent timestep t”, could you elaborate how to do this?*
>
> We apologize for the previous lack of clarity; thank you for asking for this elaboration.
>
> The procedure for augmenting the motion-unit parameters with the latent timestep $t$ is straightforward:
>
> 1. We first obtain the 3D translation parameters $[x, y, z]$ for a specific motion-unit (indexed by $p$) at a given latent time $t$.
> 2. We then augment this 3D spatial vector by **prepending** the latent time coordinate $t$, resulting in a 4D vector, $\Psi_{[p,t]} \in \mathbb{R}^4$, specifically $[t, x, y, z]$.
> 3. This final vector is then used to inform the subsequent 4D space modeling in HSI.

---

### Official Review · Reviewer_Yd3q · 2025-10-30

**Soundness:** 3
**Presentation:** 3
**Contribution:** 2
**Rating:** 6
**Confidence:** 4

**Summary:**

This paper proposes MotionWeaver, a multi-humanoid image-to-video framework that (1) learns identity-agnostic 4D motion tokens via the Unified-Choreography Core (UCC), (2) fuses motion and video latents in a shared 4D space using the Hyper-Scene Integrator (HSI) with depth-aware attention and Dynamic Cross-RoPE, and (3) trains with Hierarchical-4D Supervision (H4S) that adds occlusion supervision at high noise and motion-level supervision at low noise. The authors also curate MultiHuman46 (46 hours of multi-human video) and release DualDynamics (300 two-character clips) to stress-test interactions/occlusions, reporting consistent SOTA on that benchmark.

**Strengths:**

1. The paper clearly argues that 2D control entangles appearance and motion and lacks explicit depth reasoning in multi-person scenes, and responds with a fully 4D-anchored pipeline (UCC, HSI and H4S) that separates motion from morphology, injects depth ordering, and supervises motion explicitly.

2. UCC standardizes SMPL joints to strip appearance cues and binds per-person motion with group attention. HSI adds depth-aware cross-attention plus Dynamic Cross-RoPE to align (t, x, y) between camera and image planes. H4S schedules occlusion vs. motion-level supervision by noise step. These pieces are concrete and novel in combination.

3. This paper provides extensive experiments to support its claim. On DualDynamics, MotionWeaver tops all baselines on every metric, supporting the claim that explicit depth/occlusion handling and identity-motion binding help in multi-humanoid settings. Module-wise ablations and attention visualizations show each component (motion normalization, group attention, depth-aware attention, Dynamic Cross-RoPE, timestep-aware training) matters. Qualitative figures highlight fewer identity swaps and cleaner occlusion ordering than baselines.

**Weaknesses:**

My major concern about this paper is the detailed comparison over existing methods to show the contribution clearly. The multi-person interaction and 4D motion tokens are adopted by previous methods already. I hope the authors could clearly clarify the difference with existing methods.

1. MTVCrafter (Ding et al., 2025) also models raw 4D motion via a tokenizer (4DMoT) and conditions a DiT with 4D positional encodings, reporting large gains on open-world human animation. MotionWeaver likewise trains a 4D tokenizer, uses 4D positional cues (Dynamic Cross-RoPE), and claims generalization beyond humans, so the novelty boundary feels blurred. The paper would benefit from a direct experimental comparison (same control signals/base model), or at least an ablation contrasting quantized motion tokens (MTVCrafter) versus the authors’ standardized-skeleton motion units and group-attention binding.

2. DanceTogether (Chen et al., 2025) introduces a MaskPoseAdapter that fuses tracking masks with noisy pose heatmaps at every denoising step to suppress identity drift when actors swap positions and interact over long horizons. MotionWeaver’s identity binding relies on group attention and occlusion-aware depth cues but evaluates on 49-frame clips. It doesn’t stress identity under frequent cross-overs or long sequences. A head-to-head on DanceTogether’s scenarios/metrics, or adding long-horizon tests with frequent position exchanges, would strengthen claims. Besides, Structural Video Diffusion [a] in ICCV 2025 proposes identity-specific embeddings plus structural learning with depth + surface normals. MotionWeaver also models depth (via depth-aware attention and occlusion loss). A comparison on the [a]'s setup or cross-evaluation across datasets would clarify when explicit identity embeddings vs. identity-agnostic binding are preferable.

[a] Zhenzhi Wang, Yixuan Li, Yanhong Zeng, Yuwei Guo, Dahua Lin, Tianfan Xue, Bo Dai. Multi-identity Human Image Animation with Structural Video Diffusion, ICCV 2025. https://openaccess.thecvf.com/content/ICCV2025/papers/Wang_Multi-identity_Human_Image_Animation_with_Structural_Video_Diffusion_ICCV_2025_paper.pdf

3. DualDynamics emphasizes two-character, 49-frame interactions crafted by an animation team. MultiHuman46 includes AI-generated material and web-crawled clips. While appropriate for controlled studies, this may underestimate long-range identity drift and real-world messiness. Evaluating on longer, real-capture sequences (or adopting external multi-human sets) would improve external validity.

**Questions:**

Please see the weaknesses.

---

> ### Author Response · Authors · 2025-11-20
> **Response to Weakness 1**
>
> First of all, we are extremely grateful for the time and effort you devoted to helping us solidify this paper. We confirm that all additional discussions and experimental results will be integrated into the final paper. We hope our responses meet your satisfaction.
>
> > Weakness 1: Lacking detailed comparison with MTVCrafter (4D motion tokens)
>
> Thank you for pointing this out and giving us the opportunity to clarify.
>
> ## *Quick Summary of the Response*
>
> Our MotionWeaver fundamentally differs from and significantly expands upon MTVCrafter.
>
> - MTVCrafter is restricted to single-human. MotionWeaver is the first to pioneer **multi-humanoid** animation.
> - MotionWeaver introduces a novel 4D-anchored framework, incorporating several **essential designs** for multi-humanoid animation **that are absent in MTVCrafter**.
> - While both methods are based on 4D motion, our UCC further employs explicit decoupling mechanisms for diverse humanoid forms.
> - We employ a completely distinct 4D positional encoding scheme, where our **dynamic** design proves crucial for supporting multi-humanoid scenes.
> - Qualitative comparisons on DualDynamic (Fig. 5, page 17) show MotionWeaver's superior multi-humanoid capability, contrasting MTVCrafter's single-human limitation.
>
>
>
>
> ## *Methodological Level*
>
> - **Expanded Scope & Design**
>
>   - MTVCrafter is **inherently restricted to single-human** animation due to its underlying methodological constraints, while our proposed method fundamentally expands the scope to **multi-humanoid** image animation by our novel holistic 4D-anchored framework.
>   - MotionWeaver introduces several novel and **essential** designs for **multi**-humanoid animation that are **absent in MTVCrafter**. Specifically, these include:
>     - the binding mechanism in UCC
>     - the 4D shared Q/K space in HSI
>     - hierarchical 4D-level supervision in H4S
>     - MultiHuman46 dataset and DualDynamic benchmark.
>
> - **Enhanced 4D Motion**
>
>   - MTVCrafter directly models the **raw** 4D motion sequence, which inadvertently retains an undesirable **coupling with appearance cues**. This coupling compromises the model's generalization, occasionally resulting in the *nullification of the control signal* when the disparity between the driving signal and the driven character's appearance is substantial (see Fig. 5, row 1, page 17).
>   - Our UCC incorporates a series of **explicit decoupling mechanisms** (eg. standardized skeleton) designed to purge appearance information from the motion tokens. This strategy during training yields *exceptional robustness* during inference.
>
> - **A Paradigm Shift in 4D Positional Encoding: From Static to Dynamic**
>
>   - MTVCrafter employs an **asymmetric and static** encoding scheme [3].
>
>     - It uses 3D RoPE for video latents $(t,x,y)$ but 4D RoPE for motion tokens $(t,x,y,z)$.
>     - The 4D RoPE vectors are identical across all characters, as they stem from the entire dataset's mean positions.
>
>     This design fails to leverage the inherent strengths of the RoPE mechanism, making the positional information difficult to learn and **lacking scalability to multi-character scenarios**.
>
>   - We propose a **uniform and dynamic** design.
>
>     - Complex depth information is modeled via Depth-Aware Attention.
>     - Dynamic C-RoPE is applied across $(t, x, y)$ dimensions, where each motion token's RoPE vector is dynamically determined by its current global scene position.
>
>     This approach ensures that both motion tokens and video latent tokens at the same location share similar position encodings, fully leveraging the advantages of RoPE and native supporting for multi-character generation.
>
>
>
>
>
> ## *Experimental Level*
>
> We conducted detailed qualitative experiments against MTVCrafter on the DualDynamics benchmark **(see Fig. 5, page 17)**. The results clearly indicate that **MTVCrafter is limited to single-human settings** due to its underlying methodological constraints, while our proposed **MotionWeaver** exhibits a significant advantage and superior capability in the **broader multi-humanoid setting**. For a fair comparison, we utilized the same control signals (4D motion predicted by NLF pose estimator [1]) and the same base model (Wan2.1-I2V-14B [2]).
>
> #### Reference
>
> [1] [Neural localizer fields for continuous 3d human pose and shape estimation](https://arxiv.org/abs/2407.07532)
>
> [2] [Wan: Open and Advanced Large-Scale Video Generative Models](https://arxiv.org/abs/2503.20314)
>
> [3] [MTVCrafter: 4D Motion Tokenization for Open-World Human Image Animation](https://arxiv.org/abs/2503.20314)

---

> ### Author Response · Authors · 2025-11-20
> **Response to Weakness 2**
>
> > Weakness2: Lacking detailed comparison with DanceTogether and Structural Video Diffusion (multi-person interaction).
>
> Thank you for the valuable comment.
>
> ## *Quick Summary of the Response*
>
> DanceTogether and Structural Video Diffusion are limited to multi-**human** animation because their framework rely heavily on **explicit human masks** as control signals，which significantly impedes generalization.
>
> We introduce a new task: **multi-humanoid animation**. Due to the inherent limitations of previous technical route, this task was previously unaddressable. Our holistic 4D-anchored framework pioneers a **novel and distinct technical route** to successfully tackle this challenge.
>
>
>
>
>
> ## *Detailed Discussion Regarding DanceTogether*
>
> As you pointed out, DanceTogether introduces a MaskPoseAdapter to suppress identity drift. However, the MaskPoseAdapter **heavily relies on explicit human masks** (as shown in their ablation study), which imposes a significant constraint: *it requires the driving and driven characters to maintain identical body forms*. **Consequently, its generalization is limited.** As noted in their paper, for broader applications like human-robot interaction, this method necessitates a specialized dataset and one-hour fine-tuning.
>
> Our method, however, pursues a **completely distinct technical path**. By employing novel motion representations and training from a thoroughly 4D-grounded perspective, our model develops a genuine understanding of motion dynamics, effectively addressing the complexities of multi-humanoid animation.
>
>
>
>
>
> ## *Detailed Discussion Regarding Structural Video Diffusion*
>
> As you mentioned, Structural Video Diffusion proposes identity-specific embeddings for multi-human animation. However, their design mandates **tracking human masks** to inject the ID embeddings into the corresponding spatial regions. Methodologically, this presents the same core limitation as DanceTogether: **restricted generalization and inapplicability to diverse humanoids.** In contrast, our completely distinct approach enables the model to develop a genuine understanding of motion dynamics, thereby effectively tackling multi-humanoid animation. **This provides the theoretical answer to your final question**: if the task scope is strictly limited to human figures where body discrepancies between the driving and driven videos are minimal, Structural Video Diffusion's explicit identity embeddings might be better suited. Conversely, if the task is extended to the entire humanoid domain, demanding strong generalization capabilities, our identity-agnostic binding mechanism is demonstrably more preferable.
>
> For depth modeling, Structural Video Diffusion constructs a multi-modal UNet that jointly predicts RGB, depth, and surface normals, which significantly increases training complexity. In contrast, our HSI models depth using a more lightweight and targeted approach: **depth-aware attention coupled with an occlusion loss.** This design directly addresses the multi-character occlusion challenge in an efficient way.
>
>
>
>
>
> ## *Experiment*
>
> Due to the **lack of publicly released model weights, datasets and benchmarks** from DanceTogether and Structural Video Diffusion, a direct experimental comparison could not be performed. For confirmation, please refer to their respective GitHub repositories: [DanceTogether](https://github.com/yisuanwang/DanceTog) and [Structural Video Diffusion](https://github.com/zhenzhiwang/Multi-HumanVid/issues/1).
>
> However, we have followed your suggestions regarding *"adding long-horizon tests with frequent position exchanges"* and *"cross-evaluation across datasets"* in the following section.

---

> ### Author Response · Authors · 2025-11-20
> **Response to Weakness 3**
>
> > Weakness 3: Evaluating on longer, real-capture sequences would improve external validity.
>
> That is a very helpful suggestion, thank you！
>
> ## *Quick Summary of the Response*
>
> Although our method's comprehensive superiority for multi-humanoid animation was already evidenced by the original qualitative and quantitative evaluations, we have incorporated your valuable feedback and performed additional experiments to further solidify its robustness and generalization capabilities.
>
>
>
>
>
> ## *Cross-evaluation across datasets*
>
> The existing public benchmarks in human image animation are **limited to** **TikTok** and **Fashion**, both of which are primarily designed to evaluate **single-human setting**.
>
> We conducted both qualitative (Fig. 10, page 20) and quantitative (Table 1 and Table 2) experiments on these datasets. Our approach is capable of achieving **competitive results in the single-human setting, which prior works were explicitly tailored for.** Given the long duration of minimal motion in the Fashion dataset videos, we first created a refined test set by extracting segments with significant human action changes.
>
> **Table 1**: **Single-human setting (TikTok)**, a scenario that prior methods were explicitly tailored for.
>
> |      Model      |   FVD ↓    |   FID ↓   |  PSNR ↑   |  SSIM ↑   |  LPIPS ↓  |
> | :-------------: | :--------: | :-------: | :-------: | :-------: | :-------: |
> |   MimicMotion   |   472.51   |   34.88   |   19.30   |   0.751   |   0.220   |
> |    MusePose     |   532.75   |   53.50   |   18.20   |   0.757   |   0.248   |
> |    Animate-X    |   508.63   |   32.77   |   16.71   |   0.743   |   0.285   |
> | UniAnimate-DiT  |   402.14   |   28.47   |   19.35   |   0.765   |   0.235   |
> | RealisDance-DiT |   458.81   |   34.39   |   17.55   |   0.717   |   0.261   |
> |   MTVCrafter    | **346.65** | **27.70** | **19.74** |   0.779   |   0.219   |
> |  MotionWeaver   |   391.25   |   28.49   |   19.11   | **0.782** | **0.214** |
>
>
>
>
> **Table 2: Single-human setting (Fashion)**, a scenario that prior methods were explicitly tailored for.
>
> |      Model      |   FVD ↓    |   FID ↓   |  PSNR ↑   |  SSIM ↑   |  LPIPS ↓  |
> | :-------------: | :--------: | :-------: | :-------: | :-------: | :-------: |
> |   MimicMotion   |   481.05   |   50.11   |   17.92   |   0.866   |   0.106   |
> |    MusePose     |   475.88   |   34.13   |   17.75   |   0.849   |   0.083   |
> |    Animate-X    |   544.33   |   42.28   |   17.75   |   0.842   |   0.085   |
> | UniAnimate-DiT  |   579.20   |   68.03   |   18.54   |   0.665   |   0.377   |
> | RealisDance-DiT |   439.96   | **34.05** |   18.65   |   0.844   |   0.127   |
> |   MTVCrafter    | **402.02** |   35.12   | **18.66** |   0.849   |   0.102   |
> |  MotionWeaver   |   429.34   |   34.45   |   18.21   | **0.867** | **0.082** |
>
>
>
> ## *Long-horizon real-capture sequence tests with frequent position exchanges*
>
> We present qualitative results in (Fig. 7, page 18). We observe that our method still effectively suppresses identity drift even in this challenging scenarios, which further demonstrates the robustness of our approach. The experiments utilized 81-frame video sequences, corresponding to the maximum sequence length employed during the [training of the base model](https://github.com/Wan-Video/Wan2.1/issues/54).

---

### Author Response · Authors · 2025-11-21
**Global reply**

We sincerely thank all reviewers for their thoughtful comments and constructive feedback.

We are pleased that the feedback consistently highlights strengths of our work:

- **All reviewers** acknowledged the **significance** of our work, such as reviewer dGPY writing: "The paper tackles the crucial and challenging problem of multi-character image animation."
- **Three reviewers** pointed out our strong analysis and **reasonable motivation**, such as reviewer 9egx stating that it is "well motivated and reasonable."
- **All reviewers** agreed that our proposed method is **novel**, such as reviewer Yd3q noting: "These pieces are concrete and novel in combination."
- **All reviewers** recognized the **compelling performance** of our method, such as reviewer FnGY commenting: "which outperform recent related works substantially."

In the following, we provide clear and detailed replies to all questions and concerns, and include additional experiments on more datasets to further validate our method.

While we commit to incorporating the additional experimental results into the final version of the paper, we have temporarily refrained from modifying the main text of the paper at this stage. We plan to first engage in a productive discussion with the reviewers and will subsequently integrate the full set of supplementary experiments into Page 10 of the revised paper.

---

### Author Response · Authors · 2025-11-29
**Quick Summary for Area Chair**

We acknowledge the severe impact of the information leak incident (Nov 27) and **sincerely appreciate your time in handling our submission under these circumstances.** To save your time, we present a condensed summary of our rebuttal:

### ***Rating: 8 6 6 4 $\rightarrow$ 8 8 6 6  (Completed before Nov 26)***

Crucially, our rebuttal was reviewed and successfully addressed key concerns, leading to a rating increase from 8, 6, 6, 4 to 8, 8, 6, 6, **all before the leak incident on (Nov 27)**.

Critical Timeline：

- Nov 12: Initial Rating: 8, 6, 6, 4.
  **Reviewer dGPY (Rating 4) explicitly promised to raise score if suggested experiments were completed**.
- Nov 20: Rebuttal submitted.
- Nov 21: Reviewer 9egx **raised score** (6 $\rightarrow$ 8), *confirming the main concerns were resolved.*
- Nov 26 (03:49 AM): Reviewer dGPY **raised score** (4 $\rightarrow$ 6), *confirming the concerns were resolved.*
- Nov 27 (PM): The information leak occurred (approx. 40 hours after our final score raise).

### *Summary of Initial Feedback*

*The **initial** reviews provided highly positive evaluations of our work.*

- **All reviewers** acknowledged the **significance** of our work. (dGPY: *The paper tackles the crucial and challenging problem of multi-character image animation.*)
- **Three reviewers** pointed out our strong analysis and **reasonable motivation**. (9egx: *well motivated and reasonable.*)
- **All reviewers** agreed that our proposed method is **novel**. (Yd3q: *These pieces are concrete and novel in combination.*)
- **All reviewers** recognized the **compelling performance** of our method. (FnGY: *which outperform recent related works substantially.*)


### *Summary of Concern and Rebuttal*


Reviewer dGPY (Rating 4 $\rightarrow$ 6) & Reviewer 9egx (Rating 6 $\rightarrow$ 8) & Reviewer FnGY (Rating 8, no reply yet):

- **Different Training Data Causes Unfair Comparison:** We fine-tuned all comparable baselines on our *MultiHuman46* training set. Results demonstrated that our improvements stem from our novel framework, not just the data.
- **Single-human Generalization:** We reported performance on TikTok and Fashion datasets. *MotionWeaver* remains competitive in the single-human task, a scenario that prior methods were explicitly tailored for.
- **Efficiency & Stability:** We provided inference time comparisons (proving we are not slower than baselines) and qualitative experiments showing our UCC module effectively suppresses error propagation.
- **Detailed Inquiries & Discussions:** We have offered additional explanations to clarify these particulars and incorporated the suggestions.

Reviewer Yd3q (Rating 6, no reply yet)

- **Comparisons with Additional Methods:** We conducted comprehensive comparisons regarding both methodology and experimental results with the reviewer suggested methods, further underscoring our distinct contributions.

- **Evaluation on Longer, Real-capture Sequences:** We included comparative experiments for this setting in the Appendix, which validate the robustness of our method.

> The leak incident disrupted our original revision plan. To assist you in efficiently tracking the rebuttal history, we have **retained new experiments in the Appendix to avoid breaking the indexing referenced in our responses**. We commit to fully integrating these updates after you have reviewed the rebuttal and provided your final suggestions.

### *Core Contributions of MotionWeaver*

`Pioneering Task`: *Prior methods remain confined to single-human settings.* 	MotionWeaver **pioneers the multi-humanoid animation task**, which inherently involves diverse humanoid forms, complex interactions, and frequent occlusions.

`Paradigm Shift`: *Existing approaches adhere to a 2D paradigm, treating the model as a mindless renderer.* 	We introduce a novel **Holistic 4D-Anchored Framework** where motion representations, feature fusion, and training supervision are consistently grounded in a 4D perspective. This approach fosters a **true grasp of motion dynamics** to strengthen generalization.

`Dataset`: *Current public datasets are largely confined to single-human scenarios.* 	We curate the MultiHuman46 dataset and introduce the DualDynamics benchmark.

`Performance`: Extensive experiments demonstrate the **comprehensive superiority** of MotionWeaver across diverse scenarios.

---

### Meta-Review · Area_Chair_QdgN · 2026-01-07

**Summary:**

This paper tackles an important and challenging problem, which is multi-humanoid and multi-character image animation, and the proposed 4D-anchored formulation is well-motivated, technically sound, and delivers strong results, supported by a new dataset. For rebuttal, the authors added some experiments, such as fine-tuning strong baselines on the new dataset, single-human evaluations on TikTok/Fashion, inference-time comparison, robustness to detector errors, and longer-horizon qualitative tests, and also provided detailed clarification against related works. During the discussion period, two reviewers had explicitly updated their scores ( before the information leak). Therefore, the final recommendation is Accept.

**Reviewer Concerns:**

Addressed: 1) Fairness of comparisons: The authors fine-tuned two strong baselines (UniAnimate-DiT, RealisDance-DiT) on MultiHuman46 and still observed consistent gaps on DualDynamics; 2) single-human benchmark evaluation: The authors reported additional results on TikTok and Fashion; 3) External validity and longer-horizon behavior: Added longer real-capture qualitative tests with more diverse position exchanges to stress identity drift in harder settings.

Not fully addressed: 1) The training/benchmark focus is primarily two-humanoid, while the authors state the architecture can support more; there is only limited evidence provided. 2) Direct comparison with the most recent approaches is not provided due to the code or model availability issues of those works.

**Reviewer Scores:**

Two reviewers have clearly stated that they had raised the score ( before the information leak).

---

### Decision · Program_Chairs · 2026-01-26

Accept (Poster)